# A systematic review of controlled studies of suicidal and self-harming behaviours in adolescents following bereavement by suicide

**Laura del Carpio**[1], **Sally Paul**[2], **Abigail Paterson**[1], **Susan Rasmussen**[1]*

1 School of Psychological Sciences and Health, University of Strathclyde, Glasgow, United Kingdom,
2 School of Social Work and Social Policy, University of Strathclyde, Glasgow, United Kingdom

* s.a.rasmussen@strath.ac.uk

## Abstract

### Background

Research suggests that being exposed to the suicide of others increases risk of subsequent suicidal or self-harming thoughts or behaviours. What is less clear is whether this applies to adolescents, and if the risk exceeds that following other causes of death, which has implications on suicide prevention approaches. This study aimed to systematically review the evidence on adolescent bereavement experiences by different causes to address this gap.

### Methods

A comprehensive literature search using four databases (MEDLINE, PsycInfo, Web of Science, and Embase) identified 21 studies which measured suicidal or self-harm outcomes among bereaved adolescents aged between 12 to 18 years old. The literature was screened, data was extracted using pre-piloted forms, and risk of bias was assessed using versions of the Newcastle-Ottawa Scale; a proportion of papers were double extracted and assessed for bias. The review has been registered with PROSPERO (CRD42016051125).

### Results

A narrative synthesis of the literature demonstrated divergent findings depending on the outcome being measured. Suicide bereavement appears to be strongly associated with suicide mortality among parentally bereaved youth, while self-harm or non-fatal suicide attempts (either presenting to hospital or self-reported) showed mixed evidence. Suicidal ideation was not uniquely associated with suicide bereavement. An exploration of circumstances surrounding the death, characteristics of the person who died, and characteristics of the young person across each outcome measure suggested that earlier experiences of loss, shorter timeframes following the death, and maternal death are associated with particularly elevated risk of suicidal outcomes.

**Data Availability Statement:** All relevant data are within the manuscript and its Supporting information files.

**Funding:** This work was funded by the Wellcome Trust through a Wellcome Trust PhD studentship to LdC (Ref: 203349/Z/16/Z). The funding body had no role in the study design, data collection and analysis, decision to publish, or preparation of the manuscript. URL: https://wellcome.org.

**Competing interests:** The authors have declared that no competing interests exist.

## Conclusions

Findings suggest that suicide loss is associated with subsequent suicide, and may be associated with non-fatal self-harm. A detailed account of the risk and protective factors surrounding suicide bereavement among young people is crucial to understand the pathways through which suicidal behaviours develop. Researchers, policy makers and practitioners with an interest in suicide prevention will benefit from clarity around the needs of young bereaved individuals.

## Introduction

Despite advances in our understanding of the aetiology and risk factors for suicide, knowledge about the role of suicide bereavement remains incomplete. Suicide accounts for over 800,000 deaths worldwide annually [1], and somewhere between 6 and 135 people are thought to be affected by each suicide [2–5]. This indicates that vast numbers of people are grieving the loss of a friend, family member or acquaintance. A recent meta-analysis [6] found that approximately one in five individuals are exposed to suicide during their lifetime, and one in 20 within the past year. Similar prevalence rates were document among adolescents compared to adults.

Any bereavement, i.e. the experience of losing someone significant to death [7], can have a significant and enduring effect on young people. Adolescence is marked by a period of psychosocial, neurodevelopmental, and biological transitions. Coinciding with this is an increased vulnerability to poor outcomes such as mental health problems and risk taking behaviours [8,9]. Research has shown that suicidal or self-harming thoughts and behaviours (SSHTBs), irrespective of intent, are prevalent among young people [10,11]. This is especially concerning given that self-harm is a key predictor of completed suicide [11]. The additional trauma of a bereavement during this life stage may result in further difficulties that complicate this vulnerability.

Adolescents bereaved by any cause may experience a wide range of negative mental health outcomes including anxiety, depression, and post-traumatic stress disorder [PTSD; 12]. Also reported in the literature are increased health risk behaviours, functional impairment, perceived lack of control [13], lower peer attachment [14], and an elevated risk of SSHTBs [15]. It has been suggested that the risk of SSHTBs is especially high following a death by suicide [16,17]. However, whether such bereavement confers higher risk than other losses is yet unclear.

In an early review comparing the reactions of those bereaved by suicide to other deaths, Sveen and Walby [18] assessed 41 studies across all age groups, and concluded that people bereaved by suicide reported higher levels of rejection, shame, stigma, concealing the cause of death, and blame. No significant differences were observed with regards to general mental health, depression, anxiety, symptoms of PTSD, and suicidal behaviour (although only five studies across all age groups measured suicidal ideation and attempts). More recently, Pitman et al. [16] evaluated studies of bereavement by suicide versus other types of death. With regards to suicide risk, evidence of increased risk was found among adults bereaved by the suicide of a partner or ex-partner, and mothers bereaved of an adult child, compared to other causes of death. Preliminary evidence was also found of increased suicidal ideation and attempts among twins who had lost a co-twin. However, these findings were for the most part based on adult samples, so results are not necessarily relevant to younger age groups.

Reviewing evidence of the experiences of adolescents exposed to suicide loss, Kuramoto et al. [19] looked at studies of child and adolescent offspring exposed to parental suicide. The scant literature at the time identified only nine studies with inconsistent evidence regarding psychiatric and psychosocial outcomes, and only four considered SSHTBs as a measured outcome. Andriessen et al. [20] reported mixed findings regarding suicidal behaviours following the death of a friend (with cross-sectional studies more likely to find an association than longitudinal research), while there was compelling evidence of increased suicide risk following familial suicides. However, this review was not limited to control group studies. Hua et al. [21] examined the relationship between external cause parental deaths in childhood and suicidal behaviours in adulthood. Across 26 studies, there was overwhelming evidence of an association between childhood exposure to suicide and subsequent suicide risk, and four controlled studies suggested the risk was higher than in those bereaved by other external causes. However, this review examined outcomes which occurred in adulthood, following parental death only, and also included uncontrolled studies. More recently, Hill et al. [22] reviewed studies on exposure to suicide and suicide attempts, and found that exposure to suicide was associated with an increased odds of suicide and suicide attempts, though not suicidal ideation. Age was not found to moderate risk, as shown through comparisons of studies with a majority of youth (aged 25 or under) compared to adult participants. However, this meta-analysis excluded studies where control groups were comprised of participants who had been exposed to other modes of death. Research using bereaved control groups is essential to determine whether suicide bereaved young people have unique experiences which warrant specific supports, over and above those following any bereavement.

## Objective

While previous reviews catalogue the developments in this area, an updated literature review comprising control group studies that focus specifically on adolescents is needed. This is important given that gaps remain in our knowledge about adolescent experiences, as findings from adult studies may not translate to young people; further knowledge is also needed about which factors help explain this association, if one exists. Focusing on control group studies is necessary to clarify whether suicide bereavement specifically confers greater risk of SSHTBs compared to other forms of loss. Studies using solely non-bereaved controls limit conclusions that can be drawn about whether the experiences of a suicide-bereaved group are unique to this type of loss rather than bereavement experiences generally. Such findings would assist in the development of interventions for people bereaved by suicide (i.e. postvention), as evidence for its effectiveness is still being established [23,24]. The results of this review could also inform theoretical frameworks which highlight the pernicious effects of exposure to the self-injurious behaviour of others [e.g. 25–27]. Suicide prevention policies now recognise bereaved individuals as a key target for further research and intervention [28,29], and the vulnerability of young people has also been acknowledged. Despite this, our understanding of the role of bereavement by, or exposure to, suicide in the development of youth suicidal behaviours is still unclear. Although differences between exposure to a death and bereavement are noted, these terms may be used interchangeably in this review given the inconsistent usage in the literature.

The objective of this review was to explore the evidence on whether bereavement by suicide confers greater risk of self-harm or suicidal outcomes (thoughts and behaviours) relative to other modes of death, among adolescent groups. The following three key questions were posed:

1. Does bereavement by suicide lead to greater risk of suicidal or self-harming thoughts and behaviours in adolescents compared to bereavement by other causes?

2. Do other factors related to the death affect the relationship between suicide bereavement and adverse outcomes?

3. Finally, which measures have been used to capture outcomes in the literature?

## Methods

### Protocol and registration

A systematic approach to review the literature was used, based on the Preferred Reporting Items for Systematic Reviews and Meta-Analyses (PRISMA) guidelines [30]. The review protocol was registered in advance with PROSPERO 2016 CRD42016051125 (S1 File; accessible at: http://www.crd.york.ac.uk/PROSPERO/display_record.php?ID=CRD42016051125).

### Eligibility criteria

Criteria for inclusion comprised original empirical studies which reported on experiences of individuals exposed to or bereaved by a death, regardless of their relationship to the deceased (e.g. family members, friends, significant others). Eligible papers needed to be published in a peer-reviewed journal, in English, using any study design, and comparing exposure to suicide deaths to at least one other non-suicide bereaved comparison or control group. Studies needed to include at least one measure of self-harm or suicidal thoughts or behaviours, irrespective of intent. They also needed to include adolescent participants, who were defined as being between the ages of 12–18 years old at the time of assessment of outcomes (though the time of death was not restricted to having occurred within this age range).

Studies were excluded if they were review articles, used solely non-bereaved comparators or controls as the comparison group, or investigated deaths reported by the media or assisted suicides. Exclusions were also made where group membership was unclear, such as papers providing data on outcomes following a suicide death and any death generally (i.e. not clearly a separate, mutually exclusive cause of death group). Studies were also excluded where they focused solely on parents whose offspring had died, as it was presumed that any individuals in the suicide-bereaved group were unlikely to be 18 years old as well as the parent of a child who died by suicide, given that suicide rarely occurs in people under the age of 15, and is generally not recorded in national statistics under the age of 10 [11]. Lastly, we excluded publications which focused solely on the experience of individuals aged 18+ (inclusive of 18 year olds), where it was evident that these studies were investigating the experience of adult participants (often with a mean age significantly older than adolescent groups).

### Information sources and search strategy

A search of four electronic databases (Ovid Medline, PsycInfo, Web of Science, and Embase) was conducted on 27 May, 2019, with an update for additional publications searched in May 2020. No restrictions were applied at the search phase for date, type of study, or language. The following search terms covering the exposure, outcomes, and population of interest were used: (bereave* OR grie* OR mourn* OR loss OR survivor*) AND (suicid* OR parasuicid* OR self-harm* OR self harm* OR self-injur* OR self injur* OR self-mutilat* OR self mutilat* OR self-cut* OR self cut* OR self-immolat* OR self immolat* OR self-destruct* OR self destruct* OR self-inflict* OR self inflict* OR self-poison* OR self poison* OR overdos* OR DSH OR NSSI) AND (adolescen* OR teen* OR youth* OR young* OR child*). Reference lists of relevant papers and reviews and core suicidology journals were also searched.

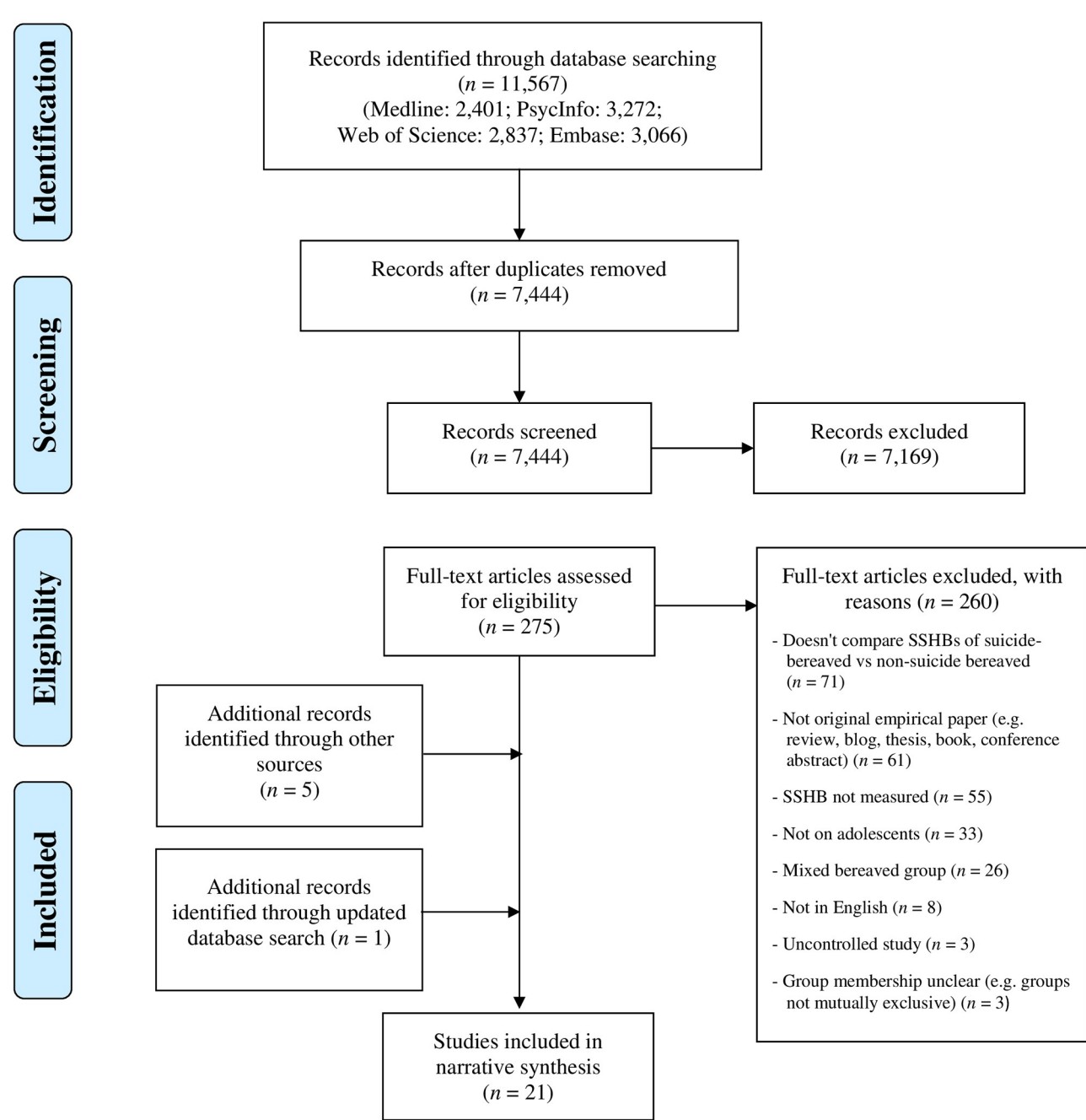

**Fig 1. PRISMA flow diagram illustrating study selection process.**

## Study selection

Literature searches, removal of duplicate references, and screening of titles and/or abstracts to identify potential papers were completed by the first author (LdC). Full text versions of papers were obtained and assessed for eligibility, with difficult cases referred to the second reviewer (SR) for discussion, and unresolved cases discussed with the third reviewer (SP). Reasons for rejections are presented in Fig 1.

## Data collection process and data extraction

A pre-piloted form was used to extract pertinent information from studies which met inclusion criteria. This covered: authors, year of publication, study aims, setting, design and procedure, participant characteristics, details of the deaths (e.g. cause, kinship relationship), outcome measure, and relevant covariates, results, and limitations. The fourth reviewer (AP) independently extracted data from over 40% of the papers ($n$ = 9) to check for accuracy, and any disagreements were resolved through discussion.

## Risk of bias

Risk of bias in individual studies was assessed by the first author using an adapted, pre-piloted version of the validated Newcastle-Ottawa Quality Assessment Scales [NOS; 31] for case-control and cohort studies. Studies were rated in three domains: selection of groups, comparability of groups, and ascertainment of exposure or outcome of interest. Quality ratings score a maximum of nine stars across the three categories, with higher scores indicating higher quality. Cross-sectional studies were assessed with a scale similar to that developed by Herzog et al. [32], based on the original NOS coding manual for cohort studies; papers were similarly rated on selection, comparability, or outcome, with a maximum possible total score of 10.

To avoid bias, a proportion of papers ($n$ = 9; 42.86%) were independently quality assessed by AP. Cohen's kappa indicated substantial agreement between ratings across individual NOS items, $\kappa$ = .864, $p < .001$. Disagreements over the risk of bias in individual papers were resolved through discussion. A decision was made to not exclude studies on the basis of their quality rating, but to incorporate the assessment of bias into the interpretation of results.

## Planned methods of analysis

A narrative synthesis of studies was planned, given the substantial heterogeneity in sample populations, exposures and outcomes measured, research designs and quality, and statistical methods across studies, which deemed such an analysis more appropriate. The primary outcome of interest was self-harming or suicidal thoughts or behaviours, irrespective of intent to die. The synthesis was structured around the outcome measured, categorised as: suicide deaths, hospitalisations for self-harm or suicide attempt, and self-reported ideation or behaviours. To address RQ1 and 2, details of each study were reported under the broad domains of: bereavement circumstances, characteristics of the person who died, and characteristics of the bereaved individual. Discussion of the methods used to capture the outcomes in the literature (RQ3) are provided within the descriptions of each study.

# Results

## Study selection

A total of 11,567 records were identified from the initial search, which was reduced to 7,444 after removal of duplicates. Titles and/or abstracts were screened for inclusion, resulting in 275 records to be fully assessed. This identified 15 includable studies, with reasons for rejections detailed in Fig 1. A further five papers were identified through hand-searching reference lists of relevant papers and reviews as well as core suicidology journals, and one additional paper was found through the May 2020 search update. A flow chart of this process is presented in Fig 1.

## Study characteristics

Details of the 21 papers selected for inclusion in this review are summarised in Table 1. The papers were published between 1992 and 2020, and some were publications of the same research groups reporting on the same sample (marked in table with superscript letters). The research took place across nine countries, mostly within Europe ($n = 13$) and the United States ($n = 6$), as well as Australia ($n = 1$) and Taiwan ($n = 1$). The study designs included nine case-control studies, eight cohort studies, and four cross-sectional studies. No qualitative studies which met inclusion criteria were identified.

Fourteen studies utilised data from national longitudinal registers which retrieved information from the whole or majority of the population over years or decades; most of these were based in European countries (Denmark, Sweden, Norway, Finland), with one in Taiwan. Smaller studies tended to be cross-sectional in design, which ranged from having 16 [49] to 40 [47] participants who had lost someone to suicide. One study recruited individuals aged 17 or above, while the remainder measured SSHTB outcomes at varying younger ages as well. The number of male participants (within bereaved groups or the total sample, as reported) ranged from 19.4% to 77.8%. Retrospective studies of people who died by suicide (mostly case-control) were overrepresented by males, while those looking at non-fatal suicide attempts included more females, reflecting the gender distribution of deaths and attempts in the general population [53,54]. Cohort studies investigating outcomes following a death generally had an equal gender distribution, while cross-sectional research which tended to employ convenience samples had a greater proportion of females.

A majority of studies looked specifically at the death of a parent ($n = 16$), while one looked only at sibling death, and the remainder at a combination of family members, friends, or acquaintances ($n = 4$). Comparison groups included bereavement by other external causes, sudden deaths, natural vs. unnatural deaths, or studies of deaths by any cause across the entire population. Outcomes were broken down into: deaths by suicide (from mortality statistics; $n = 9$), hospitalisations for self-harm or suicide attempt (from clinical registers; $n = 7$), and self-reported ideation or behaviours (assessed using interviews or self-report measures; $n = 7$; N.B., some studies measured multiple outcomes so counts do not add up to 21).

## Risk of bias

A summary of the risk of bias ratings is presented in Table 2. Ratings in the selection category were highest among case-control and cohort studies, given that the majority were based on nationwide data with large, representative samples and low risk of bias. Scores on comparability were mixed across studies; most controlled for at least one potentially confounding variable, though several did not control for depression or depressive symptoms, which was identified as the most important factor for the risk of bias assessment given its strong association with SSHTBs [53]. It is worth noting that one study which was awarded an additional star for controlling for depression actually controlled for "psychiatric history" [41], which was assumed to include depression. Cross-sectional studies scored lowest in the assessment of exposure or outcomes as they were often based on self-reports which yielded lower ratings; again, case-control and cohort studies scored highly in this category given their use of record linkage with long follow-up periods and low risk of attrition.

## Synthesis of results

The included studies suggest mixed evidence in relation to the increased risk of SSHTBs among adolescents bereaved by suicide compared to other deaths, depending on the outcome being measured. Findings are presented separately for: deaths by suicide, hospitalisations for

**Table 1. Summary of studies included in the review.**

| Citation & Country | Study Design | Participant Characteristics | Kinship Relation | Comparison Group | SSHTB Measure | Relevant Findings |
|---|---|---|---|---|---|---|
| Agerbo et al., 2002 [33] Denmark | population-based nested case-control study | 10–21 years, died between 1981–1997 77.82% male | parent or sibling | Cases: died by suicide ($n = 496$) Controls: randomly selected alive person, matched on sex, age, and time (50:1, $n = 24,800$) Bereavement by suicide ($n = 20$ cases, $n = 220$ controls), other causes ($n = 26$ cases, $n = 844$ controls), non-bereaved (number not given) | Danish medical register on vital statistics measured suicide deaths using ICD-8 (E950-E959) and ICD-10 (X60-X84) codes | Risk of suicide increased among offspring of paternal (IRR = 2.30, 95% CI = 1.10–4.80) and maternal (IRR = 4.75, 95% CI = 2.10–10.80) suicide, and maternal death by other cause (IRR = 2.06, 95% CI = 1.02–4.19). Sibling suicide or death by other cause not associated with suicide. |
| Andriessen, Hadzi-Pavlovic, et al., 2018 [34] Australia | cross-sectional study | 12–28 years ($M = 19.87$, $SD = 3.89$) 19.4% male | family member or friend | suicide ($n = 37$), natural ($n = 109$), other (accident, homicide, or unknown; $n = 30$) | Two items from novel Adolescent Grief Inventory measured self-harm or suicidal-ideation at one month post-loss and past month —items combined with anxiety-related items to form a single factor of "anxiety and self-harm" | Suicide bereaved group scored higher on anxiety and self-harm factor in the first month following the death (but not currently, in the last month), compared to those bereaved by natural death. |
| Brent et al., 2009[b] [35] USA | cohort study | 7–25 years suicide: 50.9% male accident: 45.5% male sudden natural: 59.5% male non-bereaved: 50.0% male Interviewed/followed-up at 9 and 21 months after the death | parent | suicide ($n = 53$), accident ($n = 44$), sudden natural (myocardial infarction, infections, other; $n = 79$), non-bereaved ($n = 168$) | Suicidal Ideation Questionnaire-JR (child) or Suicidal Ideation Questionnaire (adult) measured suicidal ideation | No significant differences between groups on levels of suicidal ideation after 21 months. |
| Burrell et al., 2017 [d] [36] Norway | nested case-control study using population-based registers | 0–65 years at time of parental bereavement Cases were 12–65 years when died by suicide ($M = 40.7$, $SD = 12.6$) 70.1% male | parent | Cases: bereaved person, died by suicide ($n = 375$) Controls: randomly selected alive bereaved person, matched on age, gender, and suicide date (20:1, $n = 7,500$) Bereavement by external causes: suicide ($n = 170$ cases, $n = 2151$ controls), transport accident ($n = 67$ cases, $n = 1827$ controls), other external causes ($n = 138$ cases, $n = 3522$ controls) | Cause of Death Register measured suicide deaths using ICD-8/9 (E95) and ICD-10 (X60-X84 and Y870) codes | Parental death by suicide significantly increased offspring suicide risk compared to parental death by transport accidents or other external causes. Increased risk when both parents died (vs one parent). No differences based on gender of deceased parent or offspring age at bereavement or suicide death (12–29 vs 30–65 year olds). |

(*Continued*)

**Table 1.** (Continued)

| Citation & Country | Study Design | Participant Characteristics | Kinship Relation | Comparison Group | SSHTB Measure | Relevant Findings |
|---|---|---|---|---|---|---|
| Burrell et al., 2018 [d] [37] Norway | nested case-control study using population-based registers | 0–64 years at time of parental bereavement (if bereaved) Cases were 11–64 years when died by suicide ($M = 39.3$, $SD = 13.6$) 73.36% male | parent | Cases: died by suicide ($n = 19,015$) Controls: randomly selected alive person, matched on date of birth, gender, and date of suicide (20:1, $n = 332,046$) Bereavement by external causes: suicide ($n = 209$ cases, $n = 1300$ controls), transport accident ($n = 94$ cases, $n = 1151$ controls), other external causes ($n = 182$ cases, $n = 2239$ controls), non-bereaved ($n = 11,966$ cases, $n = 234,135$ controls) | Cause of Death Register measured suicide deaths using ICD-8/9 (E95) and ICD-10 (X60-X84 and Y870) codes | All external cause parental deaths associated with increased suicide risk; higher risk following parental suicide than transport accidents or other external causes (OR = 2.86, OR = 1.36, and OR = 1.28, respectively). Risk of suicide following parental suicide higher among 11–24 year olds (OR = 4.18) than 25–64 year olds (OR = 2.63). Transport accidents and other external causes not associated with increased risk among 11–24 year olds. |
| Burrell et al., 2020 [38] Norway | nested case-control study using population-based registers | 0–18 years at time of parental bereavement (if bereaved) 43.6% male Born between 1970–2003; at least 10 years old at time of DSH | parent | Cases: received hospital treatment for DSH between 2008–2013 ($n = 12,526$) Controls: randomly selected person with no hospital-recorded DSH, matched on gender, date of birth, and date of DSH hospitalisation (20:1, $n = 222,362$) Bereavement by external causes: suicide ($n = 148$ cases, $n = 871$ controls), transport accident ($n = 54$ cases, $n = 592$ controls), other accident ($n = 76$ cases, $n = 517$ controls), other external causes ($n = 9$ cases, $n = 98$ controls), non-bereaved ($n = 12,239$ cases, $n = 220,284$ controls) | Norwegian Patient Register measured DSH presenting to hospital using ICD-10 (X6n, Y87) codes, and probable DSH also measured with ICD-10 codes related to poisoning/injuries and comorbid mental or behavioural problems, or other poisoning | Parental death by suicide (OR = 2.32) and other accidents (OR = 1.79) significantly increased risk of DSH hospitalisation compared to non-bereaved offspring, but not transport accidents or other external causes. Risk following parental suicide significantly higher than risk following parental transport accidents. No differences depending on parent or offspring gender. Highest risk of DSH hospitalisation in first 5 years following death. |
| Cerel et al., 1999 [39] USA | longitudinal study | 5–17 years suicide: $M = 11.7$, $SD = 3.4$ non-suicide: M/SD not reported but ns different Half male Interviewed/followed-up at 1, 6, 13, and 25 months after the death | parent | suicide ($n = 26$ from 15 families), non-suicide (cause other than suicide or homicide; $n = 332$) | 4-point suicidality scale summing endorsements of suicidal ideation, intent, plans, and attempts (not described) Children's Depression Inventory item measured suicidal ideation | No significant differences between bereaved groups across 18 comparisons, including suicidality. |

*(Continued)*

**Table 1.** (Continued)

| Citation & Country | Study Design | Participant Characteristics | Kinship Relation | Comparison Group | SSHTB Measure | Relevant Findings |
|---|---|---|---|---|---|---|
| Cheng et al., 2014 [40] Taiwan | nested case-control study using national registers | 15–19 years (born between 1978–1992, died between 1997–2007) Cases: 61.4% male Controls: 52.1% male | parent | Cases: died by suicide ($n = 500$) Controls: randomly selected alive person, matched on age and time (30:1, $n = 15,000$) Bereavement by suicide ($n = 9$ cases, $n = 45$ controls), other causes ($n = 62$ cases, $n = 947$ controls), non-bereaved ($n = 929$ cases, $n = 29,008$ controls) | Taiwan Mortality Registry measured suicide deaths using ICD-9 Clinical Modification (E950-E959) codes | Increased risk of suicide following paternal suicide (OR = 5.38), sig higher than paternal non-suicide (OR = 1.88), $p = .015$. Increased risk of suicide following maternal suicide (OR = 6.59), but *ns* higher than maternal non-suicide (OR = 1.94). Male cases more likely to have been bereaved by paternal (OR = 8.23), but not maternal suicide. Female cases more likely to have been bereaved by maternal (OR = 9.71), but not paternal suicide. |
| Christiansen et al., 2011[a] [41] Denmark | nested case-control study using population-based registers | Age at suicide attempt: males $M = 18.2$ years, females $M = 17.3$ years 21.3% male Born between 1983–1989; followed-up from age 10 until 2005 | parent | Cases: attempted suicide ($n = 3,465$) Controls: not attempted suicide, matched on age and gender (20:1, $n = 69,300$) Bereavement by suicide ($n = 26$ cases, $n = 200$ controls), other cause ($n = 228$ cases, $n = 2,788$ controls), non-bereaved ($n = 3,211$ cases, $n = 66,312$ controls) | Combined data from National Patient Register and Danish Psychiatric Central Registry measured suicide attempts presenting to hospital, using hospitalisation contact code E4 (suicide attempt), ICD-10 (S617-S619, X60-X84, T36-T60, or T65), ICD-8 (E9500-E9599) or ICD-10 (X60-X84) codes. | Both suicide (IRR = 4.71 and IRR = 2.25 for males and females, respectively) and non-suicide (IRR = 1.56 and IRR = 1.73 for males and females, respectively) parental deaths increased risk of suicide attempt compared to non-bereaved controls. |
| Guldin et al., 2015[c] [42] Denmark, Sweden, Finland | population-based cohort study using nationwide data | Born between 1968–2008 (Denmark), 1973–2006 (Sweden), or 1987–2007 (Finland); bereaved between 6-months and 18 years old 51.27% male Followed-up over 40 years, until own death, emigration, or end of study period (between 2008–2010) | parent | suicide ($n = 26,132$), accident (11,489), other ($n = 151,473$) Total exposed $n = 189,094$ Not exposed (non-bereaved) $n = 1,890,940$ (matched 1:10) | Cause of Death Register in each country measured suicide deaths using ICD-8/9 (950–959) and ICD-10 (X60-X84) codes | Risk of suicide elevated for bereaved children (any cause) compared to non-bereaved children, and remained high for at least 25 years after death. Highest for children whose parent died by suicide (IRR = 3.44); death by other causes IRR = 1.76. Risk of suicide higher in suicide-bereaved compared to accident-bereaved offspring (IRR = 1.82). Risk particularly high for boys who had lost a mother, offspring bereaved before 6 years, and first-born children. |

(*Continued*)

**Table 1.** (Continued)

| Citation & Country | Study Design | Participant Characteristics | Kinship Relation | Comparison Group | SSHTB Measure | Relevant Findings |
|---|---|---|---|---|---|---|
| Jakobsen & Christiansen, 2011[a] [43] Denmark | nested case-control study using population-based registers | 10–23 years ($M$ = 17.46, $SD$ = 2.37) 21.3% male Born between 1983–1989, last data update 2006 | parent | Cases: attempted suicide ($n$ = 3,465) Controls: not attempted suicide, matched on age and gender (20:1, $n$ = 69,300) Bereavement by suicide ($n$ = 27 cases, $n$ = 206 controls), natural ($n$ = 141 cases, $n$ = 1776 controls), accident ($n$ = 39 cases, $n$ = 366 controls), homicide/violence ($n$ = 13 cases, $n$ = 113 controls), unknown ($n$ = 35 cases, $n$ = 533 controls), non-bereaved ($n$ = 3,213 cases, $n$ = 66,306 controls) | Combined data from National Patient Register and Danish Psychiatric Central Registry measured suicide attempts presenting to hospital, using hospitalisation contact code E4 (suicide attempt), ICD-10 (S617-S619, X60-X84, T36-T60, or T65) or ICD-10 (X60-X84) codes. | All causes of parental death increased risk of suicide attempt compared to non-bereaved controls. No significant differences between causes of death, regardless of parental sex. Losing both parents increased risk of suicide. Risk of offspring suicide attempts sig increased for every time period (1 year to 5+ years). Moderate-high paternal income was a protective factor following maternal deaths. |
| Kuramoto et al., 2010[e] [44] Sweden | population-based retrospective cohort study using national registries | 0–17 years at time of parental death suicide: 51.4% male accident unmatched: 48.6% male accident matched: 48.7% male Followed-up over 30 years (between 1973–2003) | parent | suicide ($n$ = 23,447), accident ($n$ = 14,993/ 19,345 depending on matching) | National Inpatient Registry measured hospitalisation for confirmed or suspected suicide attempts using ICD-8/9 (E950-E959 and E980-E989) and ICD-10 (X60-X84 and Y10-Y34) codes | Maternal suicide associated with higher risk of offspring suicide attempt hospitalisation compared to maternal accidental death. Paternal suicide vs paternal accident deaths *ns* difference in hazard. Offspring of maternal death higher relative risk of hospitalisation than offspring of paternal death (p < .05). |
| Kuramoto et al., 2013[e] [45] Sweden | population-based retrospective cohort study using national registries | 0–24 years at time of parental death (0–5: early childhood, 6–12: later childhood, 13–17: adolescence, 18–24: young adulthood) suicide: 51.75% male unintentional injury: 51.50% male Followed-up over 30 years (between 1973–2003) | parent | suicide ($n$ = 26,096), unintentional injury ($n$ = 32,395) | National Inpatient Registry measured hospitalisation for confirmed or suspected suicide attempts using ICD-8/9 (E950-E959 and E980-E989) and ICD-10 (X60-X84 and Y10-Y34) codes | Similar trajectories for both groups, but offspring of those who died by suicide had earlier onset of suicide attempt hospitalisation than offspring of those who died by unintentional injury. Hazard of hospitalisation highest in first 2 years following parental suicide during adolescence or young adulthood, 5 years for parental suicide during later childhood, and 20 years for parental suicide during early childhood. Daughters higher risk of hospitalisation than sons in early childhood or adolescence (not later childhood or young adulthood). Parental sex *ns*. |

**Table 1.** (Continued)

| Citation & Country | Study Design | Participant Characteristics | Kinship Relation | Comparison Group | SSHTB Measure | Relevant Findings |
|---|---|---|---|---|---|---|
| Li et al., 2014[c] [46] Denmark, Sweden, Finland | population-based cohort study using nationwide data | Born between 1968–2008 (Denmark), 1973–2006 (Sweden), or 1987–2007 (Finland); bereaved between 6-months and 18 years old 51.3% male Followed-up until own death, emigration, or end of study period (between 2008–2010)—up to 42 years follow-up | parent | natural (diseases and medical conditions), unnatural (external causes including suicide) Total exposed $n$ = 189,094 | Cause of Death Register in each country measured suicide deaths using ICD-8/9 (950–959) and ICD-10 (X60-X84) codes | Elevated risk of all-cause mortality for almost all groups, but highest risk when offspring died from same cause as parent. Higher risk of suicide and intentional self-harm death among offspring who lost a parent to suicide (MRR = 2.78, p < .05) than non-suicide death (MRR = 1.57, p < .05). |
| McIntosh & Kelly, 1992 USA [47] | cross-sectional study | 17–72 years ($M$ = 27.9, SD = 11.0) 31.6% male | family member or close friend | suicide ($n$ = 40), accident ($n$ = 71), natural ($n$ = 63) | Suicidal Behaviors Questionnaire measured past suicidal ideation and attempts | No significant differences between groups on feeling like killing oneself after the death, or thinking about or attempting suicide during one's lifetime (significance tests not presented). |
| Melhem et al., 2008[b] [48] USA | population-based case-control study | 7–25 years suicide: $M$ = 13.6, SD = 3.7, 52% male; accident: $M$ = 13.1, SD = 4.1, 46% male; natural: $M$ = 13.4, SD = 3.4, 56% male; non-bereaved: $M$ = 12.9, SD = 3.2, 50% male Interviewed 9 months after the death. | parent [data on surviving caregivers not presented] | Cases: offspring bereaved by suicide ($n$ = 66), accident ($n$ = 51), sudden natural death ($n$ = 94) Controls: non-bereaved offspring, frequency matched by sex, age, and neighbourhood ($n$ = 183) | Suicidal Ideation Questionnaire-JR (child) or Suicidal Ideation Questionnaire (adult) measured suicidal ideation | Similar risk for suicidal ideation among suicide-bereaved offspring compared to offspring bereaved by other sudden deaths; post-hoc tests showed suicide loss group sig different from non-bereaved controls. |
| Niederkrotenthaler et al., 2012 [15] Sweden | matched case-control study using national registers | Died by suicide: $M$ = 22.3 (SD = 3.7), 72.4% male Attempted suicide: $M$ = 21.1 (SD = 4.4), 35.0% male Born between 1973–1983; at least 10 years old at time of completed or attempted suicide; Up to 31 years at end of follow-up (assessed from 1983–2004 (suicide deaths) or 1983–2006 (suicide attempts) | parent | Cases: died by suicide ($n$ = 1,407) or hospitalised for attempted suicide ($n$ = 17,159) Controls: randomly selected alive person matched on sex, month, year and country of birth ($\leq$10:1, $n$ = not stated) Bereavement by suicide ($n$ = 44 cases, $n$ = 127 controls), other causes ($n$ = 116 cases, $n$ = 694 controls) | Causes of Death Register measured suicide deaths, and National Patient Register measured suicide attempts, using ICD-8/9 (E950-E959 and E980-E989) and ICD-10 (X60-X84 and Y10-Y34) codes | Parental suicide associated with higher risk of offspring suicide (OR = 2.53) than parental death from other causes (OR = 1.30), as well as higher risk of offspring suicide attempt (OR = 1.75 and OR = 1.27, respectively). Risk of suicide following parental suicide increased when exposed at a younger age (0–10 vs >10), but suicide attempt risk increased when exposed at older age. |
| Pfeffer et al., 2000 [49] USA | cross-sectional study | 6–13 years suicide: $M$ = 9.5, SD = 2.4, 36% male cancer: $M$ = 10.4, SD = 1.9, 51% male | parent | suicide ($n$ = 11 families with 16 children), cancer ($n$ = 57 families with 64 children) | Children's Depression Inventory item on suicidal ideation measured suicidal ideation in the past two weeks | Approximately one third of both suicide and cancer bereaved groups reported suicidal ideation, similar to community normative sample. (significance tests not presented) |

(Continued)

**Table 1.** (Continued)

| Citation & Country | Study Design | Participant Characteristics | Kinship Relation | Comparison Group | SSHTB Measure | Relevant Findings |
|---|---|---|---|---|---|---|
| Pirelli & Jeglic, 2009 [50] USA | cross-sectional study | *M* = 20.0 years, *SD* = 3.97 25.3% male | immediate family, non-immediate family, friend, or acquaintance | suicide, chronic disease, acute disease, accident, murder Total *n* = 396 (individual group numbers not reported) | Self-report measure developed for this study, including questions on how many times participant has thought of "committing suicide or attempted suicide" ever and in the past year Beck Scale for Suicide Ideation measured suicidal ideation or attempts in the previous week | Type of death experienced did not predict suicidal ideation. Total suicide deaths experienced (positively) and acute deaths experienced (negatively) predicted history of suicide attempt. Other causes of death *ns*. Suicide of a friend (but not other relationships) predicted history of suicide attempt. |
| Wilcox et al., 2010 [51] Sweden | population-based retrospective cohort study using national registries | 0–25 years at time of parental death (0–12: childhood, 13–17: adolescence, 18–25: young adulthood) suicide: 52% male accident: 51% male other: 51% male non-bereaved: 51% male | parent | suicide (*n* = 44,397), accident (*n* = 41,467), other (*n* = 417,365), non-bereaved (3,807,867) | Cause of Death Register measured suicide deaths using ICD-8/9 (E950-E959) and ICD-10 (X60-X84) codes Hospital Discharge Register measured psychiatric hospitalisations for suicide attempt using ICD-8/9/10 codes above | Suicide-bereaved offspring at increased risk of suicide compared to offspring of alive parents (adjusted IRR = 1.9 across all ages); no increased risk among offspring of other causes of parental death compared to non-bereaved. Increased risk of suicide when offspring bereaved by suicide during childhood or adolescence (adjusted IRR = 3.0 and 3.1, respectively) compared to non-bereaved youth, but not young adulthood. All bereaved groups were at increased risk for hospitalisation for suicide attempt compared with offspring of alive parents, but offspring of those who died by suicide at greater risk (adjusted IRR = 1.7, vs accidental, IRR = 1.4, and other deaths, IRR = 1.3). |
| Yu et al., 2017 [52] Denmark, Sweden | population-based cohort study using nationwide data | Born between 1973–2004 (Denmark), or 1973–2006 (Sweden); bereaved between 6-months and 18 years old Exposed: 51.5% male *Unexposed*: *51.3% male* Followed-up until own death, emigration, or end of study period (2009 in Denmark, 2008 in Sweden) | sibling | neoplasms; endocrine, nutritional and/or metabolic diseases; diseases of the nervous system; diseases of the circulatory system; transportation accidents; suicide and intentional self-harm; other diseases Total exposed *n* = 55,818 *Total unexposed (non-bereaved): 4,949,211* | Cause of Death Register in each country measured suicide and intentional self-harm deaths using ICD-8/9 (950–959, 960–969) and ICD-10 (X60-X84, Y10-Y34) codes | Suicide or intentional self-harm mortality risk increased for individuals whose sibling died of suicide (MRR = 8.01) compared to non-bereaved, but *ns* when sibling died of other causes. |

Note: Similar superscripts are used for multiple publications reporting on the same or a subset of the same sample. Terminology (e.g. regarding SSHTBs) kept consistent with the original publication. Statistics are presented for adjusted analyses wherever possible. ICD = International Classification of Diseases. M = Mean. SD = Standard Deviation. OR = Odds Ratio. MRR = Mortality Rate Ratio. IRR = Incidence Rate Ratio. DSH = Deliberate self-harm.

**Table 2. Risk of bias assessments of included studies according to study design.**

| Citation | Selection | Comparability | Exposure/Outcome | Total |
|---|---|---|---|---|
| **Case-Control** | | | | |
| Agerbo et al., 2002 [33] | 4 | 2 | 3 | 9 |
| Burrell et al., 2017[d] [36] | 4 | 1 | 3 | 8 |
| Burrell et al., 2018[d] [37] | 4 | 1 | 3 | 8 |
| Burrell et al., 2020 [38] | 4 | 1 | 3 | 8 |
| Cheng et al., 2014 [40] | 4 | 1 | 3 | 8 |
| Christiansen et al., 2011[a] [41] | 4 | 1 | 3 | 8 |
| Jakobsen & Christiansen, 2011[a] [43] | 4 | 1 | 3 | 8 |
| Melhem et al., 2008[b] [48] | 2 | 1 | 0 | 3 |
| Niederkrotenthaler et al., 2012 [15] | 4 | 1 | 3 | 8 |
| **Cohort** | | | | |
| Brent et al., 2009[b] [35] | 3 | 0 | 2 | 5 |
| Cerel et al., 1999 [39] | 3 | 0 | 1 | 4 |
| Guldin et al., 2015[c] [42] | 4 | 1 | 3 | 8 |
| Kuramoto et al., 2010[e] [44] | 4 | 1 | 3 | 8 |
| Kuramoto et al., 2013[e] [45] | 4 | 1 | 3 | 8 |
| Li et al., 2014[c] [46] | 4 | 1 | 3 | 8 |
| Wilcox et al., 2010 [51] | 4 | 1 | 3 | 8 |
| Yu et al., 2017 [52] | 4 | 1 | 3 | 8 |
| **Cross-Sectional** | | | | |
| Andriessen, Hadzi-Pavlovic, et al., 2018 [34] | 3 | 0 | 2 | 5 |
| McIntosh & Kelly, 1992 [47] | 2 | 1 | 1 | 4 |
| Pfeffer et al., 2000 [49] | 3 | 1 | 2 | 6 |
| Pirelli & Jeglic, 2009 [50] | 1 | 0 | 2 | 3 |

Note: Similar superscripts are used for multiple publications reporting on the same or a subset of the same sample. Case-control and cohort studies score a maximum of nine stars, and cross-sectional studies a maximum of 10 stars, across selection, comparability, and ascertainment of exposure/outcome categories; higher scores indicate lower risk of bias.

self-harm or suicide attempts, and self-reported suicidal/self-harm ideation and behaviours. Within each of these three categories, evidence regarding the influence of bereavement circumstances, characteristics of the person who died, and characteristics of the bereaved individual are discussed where such evidence exists. While additional outcomes may have been presented in the literature, priority is given to reporting those which relate to the aims of this review.

## Association between bereavement and subsequent suicide

Seven papers measured suicide deaths, and two combined suicide deaths and suicide attempt hospitalisations. All were derived from population-based national registers, either case-control or cohort in design, and thus utilised Cause of Death or medical registers employing ICD-8/9/10 codes to determine suicides. As such, they obtained relatively large sample sizes and high statistical power, with low risk of bias. The majority looked at the effects of parental loss, with one [52] investigating sibling death, and one [33] both parental and sibling death.

**Bereavement characteristics.** All studies found that experiencing an immediate family member's death by suicide was associated with a significantly higher risk of suicide mortality compared to bereavement by other causes, after controlling for a range of confounding

variables. Exposure to other causes of death was usually also associated with an elevated risk of dying by suicide, but to a lesser magnitude than following a suicide loss [15,33,36,37,40,42,46].

After adjusting for parental psychiatric hospitalisations and criminal convictions, Wilcox et al. [51] found that offspring of those who died by suicide had an almost doubled risk of dying by suicide compared to non-bereaved youth (IRR = 1.9), while offspring of those who died by accident or other causes were not at elevated risk of suicide. The authors also examined suicide attempt risk, and found that all causes of parental death were associated with hospitalisations for suicide attempt, but those exposed to parental suicide (IRR = 1.7) were at higher risk compared to those exposed to accidents (adjusted IRR = 1.4) or other deaths (IRR = 1.3).

In a sample of adolescents in Taiwan, Cheng et al. [40] found a significantly higher risk of suicide following paternal suicide compared to non-suicide death (OR = 5.38 vs OR = 1.88, *p* = .015), as well as a higher but non-significant increased risk following maternal suicide compared to non-suicide death (OR = 6.59 vs OR = 1.94, *ns*). The authors controlled for birth characteristics and family socio-economic factors, but did not control for familial psychiatric disorders, which may explain the higher risk than seen in similar studies which did account for this potential confounder [33,42,51].

Some studies demonstrated a higher risk of mortality when offspring died from the same cause as their parents. Li et al. [46] looked at the impact of parental death in childhood or adolescence in three Nordic countries (Denmark, Sweden, and Finland). Losing a parent to any cause was associated with an increased all-cause mortality, but the highest risks were evident among concordant causes. The mortality risk ratio (MRR) among offspring who died by suicide was 2.78 when their parents had died by suicide, compared to 1.57 when their parents had died by non-suicide. Additionally, those exposed to parental suicide were more likely to die by any cause than people exposed to other deaths.

*Cumulative exposure to death*. A potential dose-response relationship was found such that the risk of offspring suicide was intensified when both parents died compared to only one parent [36,37]. Sub-group analyses of specific causes of death were not examined with regards to cumulative exposures.

*Time since death*. The consequences of suicide bereavement may persist in the short- and long-term. Li et al. [46] found an increased risk of offspring mortality following parental death by all causes until early or mid-adulthood, irrespective of socioeconomic differences. Burrell et al. [37] found that suicide risk was elevated from 1 year after parental death to over 15 years, but not within the first year post-death. They suggest that individuals may experience immediate difficulties from early feelings of trauma and loss, and suffer longer-term consequences as a result of losing a caregiver and an important attachment figure during a critical stage of development [37].

**Characteristics of the deceased person.** *Sex*. The evidence is somewhat contradictory regarding the role of parental sex. Burrell et al. [36] found that parental death often led to increased risk of suicide regardless of parental sex. Offspring who lost a father (IRR = 2.30, 95% CI = 1.10–4.80) or a mother (IRR = 4.75, 95% CI = 2.10–10.80) were both at elevated risk of dying by suicide, with confidence intervals overlapping. Similar findings were reported in Burrell et al. [37], and also following any parental death in other Nordic countries [46]. On the other hand, a higher risk of suicide was found among those bereaved by maternal than paternal suicide by Agerbo et al. [33]. Among a wide range of familial, psychiatric and socioeconomic risk factors, paternal death by suicide (IRR = 2.30, 95% CI = 1.10–4.80), maternal death by suicide (IRR = 4.75, 95% CI = 2.10–10.80), and maternal death by other causes (IRR = 2.06, 95% CI = 1.02–4.19) were found to be associated with an increased risk of dying by suicide, after adjusting for age, sex, calendar time, and individual and family history of admission for

mental illness. The IRR following maternal suicides was higher than paternal suicides, as well as other maternal bereavements, but the confidence intervals overlapped.

Among 15–19 year old adolescents, Cheng et al. [40] found that risk of own suicide in males was significantly associated with paternal (OR = 8.23, 95% CI = 2.96–22.90) but not maternal (OR = 3.50, 95% CI = 0.41–30.13) suicide, while risk in females was associated with maternal (OR = 9.71, 95% CI = 1.89–49.94) but not paternal (OR = 2.42, 95% CI = 0.30–19.57) suicide. However, the difference in risk after losing a father compared to a mother was not statistically significant (for both sons and daughters).

*Relationship*. Death of a parent was consistently associated with offspring suicide, and only two studies looked at the impact of a sibling death. Yu et al. [52] found that all-cause mortality was increased over the next 37 years following sibling death, irrespective of age at bereavement and type of death. This was particularly strong within the first year and among same-sex and close aged siblings. Risk of death by suicide or intentional self-harm was eight times higher for people whose sibling died by suicide (MRR = 8.01, 95% CI = 5.34–12.00) compared to non-bereaved siblings; other causes of loss were non-significant (MRR = 1.04, 95% CI = 0.76–1.43). On the other hand, Agerbo et al. [33] found that suicide risk was elevated following parental death but not sibling death, after controlling for age, sex, calendar time, and individual and family history of admission for mental illness. However, sibling psychiatric history was found to be a risk factor for suicide.

**Participant characteristics.**   *Age*. The impact of a death may differ by the developmental period during which it occurred, although there are contradictions within the literature regarding what age places a child most at risk. Wilcox et al. [51] showed that offspring suicide risk was particularly pronounced when they were bereaved by suicide during childhood or adolescence (adjusted IRR = 3.0 and 3.1, respectively) compared to those not bereaved, while young adults were not at increased risk. Li et al. [46] found that all-cause mortality was elevated for all age groups who were bereaved, although the risk of unnatural deaths shortly after the bereavement was particularly high for those who were younger (under 5) when bereaved. Guldin et al. [42] found a more than threefold risk of suicide among children whose parent died of suicide (IRR = 3.44), although it was also elevated for those bereaved by other causes (IRR = 1.76). Risk was highest among children who had been bereaved under the age of six, boys who lost a mother, and first-born children, and the elevated risk remained for at least 25 years. Niederkrotenthaler et al. [15] found that exposure to parental suicide from 0–10 years was associated with increased risk of dying by suicide, while exposure at an older age (over 10 years) was associated with increased risk of attempting suicide. Conversely, exposure to non-suicide deaths increased risk of offspring suicide and attempted suicide only for those above 10 years old, and the odds ratios for both outcomes were lower than those following parental suicide.

Among 11–64 year olds who died by suicide, Burrell et al. [37] found that suicide risk was increased for those bereaved by any external cause at all ages until 44 years. However, risk was especially high when the bereavement occurred between 10 to 17 years; OR = 2.24, 95% CI = 1.75–2.86. In a separate analysis, among those under 25 years at the time of their suicide, only parental death by suicide (not transport accidents or other external deaths) was significantly associated with suicide risk (OR = 4.18, 95% CI = 2.79–6.27), and the odds were higher than when bereaved at an older age (OR = 2.63, 95% CI = 2.18–3.18). There was no effect of offspring gender, and no interaction between gender of the deceased parent and bereaved offspring. In contrast Burrell et al. [36] found an increased risk of suicide mortality following parental suicide regardless of offspring age at bereavement (between 0–65 years), and regardless of age group of offspring at the time of their own suicide (12–29 year olds vs 30–65 year olds).

## Association between bereavement and subsequent hospitalisation for self-harm (irrespective of intent)

In addition to the two papers discussed above which measured both suicide deaths and hospital-presenting suicide attempts [15,51], five further studies investigated presentations to hospital for self-harm or suicide attempt. All were case-control or cohort in design, derived their data from national registers, and scored fairly low on risk of bias. All investigated the impact of a parent's death, and were based in Scandinavian countries (Norway, Denmark or Sweden).

**Bereavement characteristics.**   There were mixed findings in terms of hospitalisations for self-harm or suicide attempt. Several studies looked at all-cause mortality across the population, and three looked exclusively at suicide compared to other external causes of death. All controlled for a range of potential confounders from registry data, although Jakobsen and Christiansen [43] only controlled for age and sex.

Burrell et al. [38] compared records of hospitalisations for self-harm among people over 10 years old who were bereaved by suicide, external causes (transport accident, other accident, other external cause), or matched non-bereaved controls. Among individuals treated for self-harm, both those who had lost a parent to suicide (OR = 2.32, 95% CI = 1.92–2.80) and other accident (OR = 1.79, 95% CI = 1.38–2.33) had a greater risk of hospitalisation compared to non-bereaved controls, although the confidence intervals of the two bereaved groups overlapped. Transport accidents and other external causes of bereavement were not associated with an increased risk, and suicide-bereaved offspring were at significantly higher risk than those bereaved by transport accidents. Jakobsen and Christiansen [43] showed that all causes of death studied (suicide, accident, homicide, and violence) were associated with adolescent suicide attempt risk, with odds ratios ranging from 1.64 to 2.70, and there were no significant differences between groups. However, the small number of suicide attempts per each cause of parental death may account for the lack of significance in the results (e.g. only 13 people who attempted suicide were bereaved by homicide or violence, and 27 by suicide).

Christiansen et al. [41] found that parental death by any cause increased risk for offspring suicide, as did other factors such as parental history of non-fatal suicide attempts, psychiatric illness, and low income. Suicide bereavements were associated with a higher incidence rate ratio than other causes of death, both in males (IRR = 4.71 vs IRR = 1.56) and females (IRR = 2.25 vs IRR = 1.73), although the confidence intervals only overlapped for females. It is worth noting that the group sample sizes were not extensive; for instance, only eight cases who attempted suicide had a parent who died by suicide. The authors also found a cumulative effect of parental risk factors on offspring risk of suicide.

*Cumulative exposure to death.* The loss of both parents was associated with a more than threefold (OR = 3.09) increased risk of hospitalisation in Burrell et al. [38], higher than the risk following the loss of only one parent (OR = 1.76 and OR = 2.00 for fathers and mothers, respectively). In similar findings, Jakobsen and Christiansen [43] found that losing both parents was associated with a much higher risk of offspring suicide compared to losing one parent (RR = 4.66 vs 1.71). However, the sample of people bereaved by both parents was small in both studies.

*Time since death.* A common finding was that the highest risk of hospitalisation exists in the short term following the death, particularly within the first two years. Jakobsen and Christiansen [43] found that the risk was significantly higher among bereaved (combined groups) compared to non-bereaved individuals across all time periods (from up to one year to more than five years), although the highest risk was within the first year (OR = 2.06) and gradually decreased with each increasing period (e.g. OR = 1.65 at more than five years). Kuramoto et al. [45] showed that risk for suicide attempt hospitalisation was highest in the first two years

following adolescent or young adult exposure to suicide, and gradually decreased thereafter (followed-up over 30 years). The largest risk of hospitalisation for self-harm in Burrell et al.'s [38] study was seen within the first five years following the death (OR = 15.61), and remained significant until 10 years post-loss, but not from 10–15 years after.

**Characteristics of the deceased person.** *Sex.* Three studies found no effect of a deceased parent's sex on risk of hospitalisation for self-harm or suicide attempt [38,43,45]. On the other hand, Kuramoto et al. [44] found that offspring exposed to maternal suicide had an almost doubled risk of hospitalisation for suicide attempt compared to those exposed to maternal accident (Hazard Ratio = 1.80, 95% CI = 1.19–2.74), however no significant differences were found between paternal suicide or accidental deaths.

**Participant characteristics.** *Age.* No clear patterns emerged from the data regarding the impact of age on risk of hospitalisation. Wilcox et al. [51] showed that those exposed to parental suicide during childhood (0–12 years) were more likely to attempt suicide (adjusted IRR = 1.9) than adolescents or young adults (adjusted IRR = 1.6 for both). Kuramoto et al. [45] looked at time to hospitalisation for suicide attempts among people who had lost a parent prior to the age of 25. Both those who had been exposed to parental suicide or unintentional injury showed similar trajectories of risk; however, offspring of those who died by suicide had earlier onset of hospitalisation. Those bereaved in early childhood (0–5 years) showed an elevated risk that continued to increase for 20 years, those bereaved in later childhood (6–12 years) showed an increase in risk over the next five years, and those bereaved during adolescence (13–17 years) or young adulthood (18–24 years) showed the highest risk within the first 2 years after the death. These findings support the view of a critical period shortly after the death, but also that earlier bereavements have a more detrimental effect.

*Sex.* Kuramoto et al. [45] found that female offspring who had lost a parent to suicide were more likely to be hospitalised for suicide attempt than male offspring, but only when they were bereaved during early childhood or adolescence (not later childhood or young adulthood).

## Association between bereavement and subsequent self-reported suicidal ideation or self-harm (irrespective of intent)

Three studies measured suicidal ideation only, while a further four measured a combination of suicidal ideation and self-harm. As all measured self-reported outcomes, they were prone to response and recall bias, and many were cross-sectional and thus unable to make inferences about causality. Risk of bias also varied greatly between studies, and while the majority adjusted for a variety of potential confounders, four studies did not report any such adjustments [34,35,39,50].

**Bereavement characteristics.** Suicide bereavement was contrasted with a range of comparison groups including sudden deaths, natural deaths, accidents, homicide, cancer, chronic and acute diseases, or non-suicide causes more generally. Few differences were apparent in risk of suicidal ideation; only one of the seven studies [34] reported possible differences between the bereaved groups. Andriessen et al. [34] measured suicidal ideation and self-harm using two items from a novel Adolescent Grief Inventory regarding thoughts or behaviours at one month after the death, and in the past month. Compared to those bereaved by natural deaths, adolescents bereaved by suicide reported higher scores at one month post-loss on an 'anxiety and self-harm' factor (items were combined in a factor analysis as they loaded onto one factor). However this effect was non-significant for responses concerning the past month. As this study combined ideation and behaviours into one composite measure, it is not possible to determine whether ideation specifically was elevated among the suicide bereaved individuals.

Mixed results were found regarding self-reported self-harm or suicide attempts. Two papers found no significant differences between bereaved groups [39,47], while another two found that suicide bereaved young people were more likely to endorse a history of self-harm or suicide attempts [34,50]. As noted above, Andriessen et al. [34] reported that suicide-bereaved adolescents scored higher on an anxiety and self-harm factor in the first month following the death than the natural death bereaved group, although the composite measure used precludes conclusions on specific risk of behaviours rather than thoughts. Pirelli and Jeglic [50] measured both thoughts of suicide and suicide attempts using a self-generated measure and the Beck Scale for Suicide Ideation [55]. Although they did not control for potential confounding (and risk of bias was deemed to be high), results showed no differences in suicidal ideation between suicide and other bereaved groups (chronic disease, acute disease, accident, and murder; group sizes not reported). However, those reporting a suicide attempt were more likely to have been bereaved by suicide and less likely bereaved by acute disease. Furthermore, it was the suicide of a friend, and not a family member or acquaintance, which accounted for this relationship.

*Time since death*. Time since death varied from 1 month to up to 34 years, with studies of shorter time periods (i.e. within the past two years) less likely to find group differences than those measuring longer periods. Two studies were longitudinal. Cerel et al. [39] compared grief reactions, psychiatric symptomatology, and psychosocial functioning of offspring at 1, 6, 13, and 25 months following the death. Findings revealed no significant differences regarding suicidality (measured by suicidal ideation, intent, plans, and attempts), as well as several other measures (e.g. posttraumatic stress symptoms, depressive symptoms, psychosocial functioning, etc.), although the suicide bereaved group were more likely to experience anxiety, anger, and shame, and less relief and acceptance. However, the risk of bias in this study was high, and it is worth highlighting the small (and perhaps unrepresentative) sample, the use of unvalidated measures, and a large number of statistical analyses that did not control for potential confounders or correct for multiple comparisons.

Melhem et al. [48] looked at families with offspring between 7–25 years in which one parent had died by suicide, accident, or sudden natural death, as well as a control group of matched, non-bereaved offspring. Participants were interviewed nine months after the death, and completed versions of the Suicidal Ideation Questionnaire [56,57]. After controlling for demographic, familial and clinical risk factors, suicide bereaved offspring were found to show a similar risk for suicidal ideation compared to other sudden deaths ($p = .02$, where α was significant at .008). These findings suggest that in the short term, outcomes may be similar for suicide and other sudden death bereaved offspring. However, the sample size was small and possible selection bias was reflected in the high risk of bias rating. In a follow-up of this cohort at 21 months, Brent et al. [35] continued to find no significant differences between groups on the measure of suicidal ideation, although differences were noted for other outcomes (e.g. depression, substance use disorder, etc.).

**Characteristics of the deceased person.**   *Sex*. No studies measuring self-reported outcomes found significant gender differences, although such analyses were not always conducted or reported. Although they did not find significant group differences in relation to suicidal ideation between different bereaved groups, Brent et al. [35] reported that depression between nine and 21 months post-loss was associated with the death of a mother rather than a father.

*Relationship*. Four studies investigated the death of a parent, and three looked at a combination of family members, friends or acquaintances. None of those looking at parental death reported significant differences between suicide and other bereaved groups in terms of suicidal ideation [35,48,49], nor ideation or attempts [39]. Among those investigating other relationships, Pirelli and Jeglic [50] only found an association with suicide attempts (and not suicidal

ideation) among friends, and not family members or acquaintances. Andriessen et al. [34] found that those who were not blood-related to the person who died had significantly higher scores on an anxiety-self-harm factor compared to other family members (not first-degree relatives) in the first month after the death, though no differences were observed when asked about the past month. McIntosh and Kelly [47] did not report comparisons between those bereaved by family members or friends.

**Participant characteristics.** *Age*. Five studies included individuals with a broad age span that covered the adolescent years (12–18 years), while two focused on participants who were slightly older comprising university student samples [47,50]. No clear age differences were apparent. Among the younger age groups, Pfeffer et al. [49] examined prepubescent children and young adolescents aged 6–13, and found similar rates of suicidal ideation between suicide and cancer bereaved children. However, the sample size was small (*n* = 16 in the suicide group and *n* = 64 in the cancer group), and significance tests were not presented. Other studies with younger samples also reported no significant differences among groups.

Andriessen et al. [34] found differences in an anxiety and self-harm factor (with those bereaved by suicide more likely to exhibit higher scores compared to those bereaved by natural deaths) among participants aged 12–28, although the mean age in this study was 19.87 years. Although they did not specify the age range of their sample, Pirelli and Jeglic [50] reported data from participants with a mean age of 20 years (SD = 3.97), with their sample concluding mixed findings (no differences were found in relation to suicidal ideation, but suicide attempts were more common among those bereaved by suicide compared to acute disease).

Across older ages, McIntosh and Kelly [47] reported results using the Suicidal Behaviors Questionnaire [58], which measured past suicidal ideation and attempts. No differences were noted between bereaved groups on this measure, although significance tests were not presented. Furthermore, the sample was not necessarily representative, the overall closeness of the relationships were rated as moderate so may not reflect reactions to the loss of a more intimate relationship, and participants recalled deaths from an unlimited period of time ago (up to 34 years, despite controlling for this variable), suggesting possible recall bias.

## Discussion

### Summary of findings

This review aimed to examine the literature regarding risk of SSHTBs among adolescents who have experienced a death. The aims were to: (1) determine whether suicide bereavement leads to an elevated risk of SSHTBs compared to other modes of death, (2) to examine factors which may help explain this potential relationship, and (3) to explore the measures used to ascertain outcomes in the literature. Following a systematic approach, 21 papers met inclusion criteria and provided evidence regarding outcomes following bereavement. Overall, the evidence was robust as to the increased risk of suicide mortality following parental suicide, and tentative evidence was found of increased risk following sibling suicide. Hospitalisations for self-harm were somewhat more common among some suicide bereaved adolescents compared to other groups, but there were contradictory findings. With self-reported outcomes, no differences were apparent in risk of suicidal ideation, while self-reported behaviours showed mixed evidence between those bereaved by suicide or other deaths.

These findings are largely consistent with conclusions by Hill et al. [22] in their recent review, which indicated an elevated risk of suicide and suicide attempts but not suicidal ideation among those exposed to suicide. We found that several factors were identified as being relevant to adjustment following a death. Cumulative exposures to death were consistently associated with more negative outcomes. The impact of a double loss for adolescents may lead

to additional changes in childcare routines or residence, which complicate adjustment after the loss [36]. On the whole, experiencing death at a younger age (especially in childhood) seems to be more detrimental, which supports the idea of sensitive life periods, with increased vulnerability coinciding with puberty, biopsychosocial changes, and higher impulsivity and emotional immaturity [15,37,59]. This also points to an environmental or developmental component to suicide risk, rather than just genetic [51], and highlights the importance of social and relational circumstances in dealing with bereavement and possibilities for early intervention. Outcomes measured soon after the death, particularly within the first two years, appear to be most deleterious, although some studies showed that risk of SSHTBs persisted for decades. Inconsistent evidence was found regarding the effect of parental sex, with a small number of studies noting a more detrimental effect of maternal loss [e.g. 33,44], while one study found an elevated risk among same-sex parent-child dyads [40]. This is consistent with review findings by Geulayov et al. [60], suggesting that maternal suicidal behaviour and younger exposure to parental suicidal behaviour are associated with greater offspring risk. This gender effect may be attributed to losing the parent that is generally the primary caregiver [44], while Cheng et al. [40] suggest that the death of a same-sex parent may represent the loss of a stronger attachment figure and role model from which to learn coping skills, which may include self-harm. In terms of kinship, loss of a parent, and often a sibling, was overwhelmingly associated with poor outcomes, while studies of non-familial relationships were inconclusive.

## Possible mechanisms

The increased risk of suicide among adolescents bereaved by a parent or sibling may be explained by familial clustering based on shared genetic vulnerabilities for suicidal behaviour [36,42,61]. This may include transmission of mental health problems, neurocognitive deficits, and personality traits reported to influence suicidal behaviour [38]. Brent and Melhem [62] noted that traits such as impulsive aggression may run in families, and adverse family environments (such as those involving abuse and maltreatment) preceding the death may similarly contribute to familial aggregation of suicide.

Significant associations were not only seen for familial bereavements. Andriessen et al. [34] found that those bereaved by a family member or friend's suicide scored higher on a factor measuring anxiety and self-harm than those bereaved by other causes, and Pirelli and Jeglic [50] found that only a friend's suicide (rather than an immediate or non-immediate family member or acquaintance) was predictive of post-loss suicide attempts. Peer influence on risk of suicidal thoughts and intentions may be explained by assortative relating, with vulnerable adolescents more likely to socialise with fellow peers who are similarly at risk, and their increased vulnerability is exacerbated by shared life stresses [63]. Research on social modelling of behaviours applies to both familial and peer relationships [64]. It may be that suicide becomes a learned coping style and viable option to deal with stress after witnessing the suicidal behaviour of others [65], and the concept of modelling applies to behaviours rather than internal processes.

In their recent paper reviewing evidence of clustering of suicides in childhood and adolescence, Hawton et al. [66] propose that, in addition to social transmission and assortative relating, descriptive norms (perceptions of suicidal behaviour being widespread) and social cohesion (especially related to shared ideas and attitudes) are likely to account for this effect. Lastly, the stigma surrounding suicide, applicable to both familial and non-familial bereavements [17,18], may explain why some suicide-bereaved individuals show poorer outcomes, and may be reflected in a reluctance to seek help when needed (compounded by a lack of information about help seeking and access to support more broadly). Hua et al. [67] reviewed

studies that employed a theory to explain the relationship between childhood parental bereavement and subsequent suicidal behaviour in adulthood. Overall, the theories used tend to cover a range of factors, where in actuality, it is likely that multiple biopsychosocial factors interact to influence suicide risk following bereavement.

## Measures

There was wide variability across the literature in terms of measures used to capture SSHTB outcomes. Data from national registers, which measured both suicide deaths and hospitalisations for self-harm or suicide attempt, offered vast amounts of data with high validity and statistical power, and the opportunity to follow-up individuals over lengthy periods. Research using population registers allows for the exploration of rare exposures such as suicide, and the data are truly representative of the population in those countries, thus posing limited risk of selection bias [68]. Potential inaccuracies in death registrations may however underestimate the number of people who die by suicide, and given that suicide has a low base rate, some analyses (e.g. stratified by sex or other factors) may still be underpowered. Register data is also limited in information on family environment, such as whether the individuals in question were residing together, and other potential factors that intensify or attenuate the relationship between bereavement and SSHTBs. For instance, circumstances of the death, the quality and closeness of the relationship, the remaining social supports, and individuals' coping mechanisms may all be important but not captured in routine register data [36]. It is also worth noting that research measuring hospitalisations for self-harm only capture the more severe cases which receive clinical attention. As evidence consistently shows, community-occurring self-harm is particularly prevalent among young people [69], so more high quality studies are needed with adolescents not presenting to clinical settings.

Research reporting self-reported outcomes (ideation and self-harm) relied on data from interviews or questionnaires, and suffered from the common pitfalls of self-report (e.g. recall bias or socially desirable responding). Many of these studies used validated tools, but a few used unvalidated measures or did not describe the measures appropriately. Several studies also failed to adjust for multiple comparisons or potential confounding, and given their generally small sample sizes with low statistical power, analytical approaches were at times questionable. Unlike register-based studies which assessed the sequelae of immediate familial bereavement, studies on self-reported outcomes examined mixed relationships such as peer and non-immediate familial deaths. This may explain the diverging findings regarding increased risks of self-harm, and further research with larger representative samples is necessary to explore this further. Finally, several self-report studies tended to be cross-sectional in design, so conclusions about causality are limited.

## Strengths and limitations

This systematic review explored the literature on the experiences of adolescents who have been bereaved by suicide compared to other causes. Previous reviews took a broader focus in terms of exposure (e.g. including non-bereaved controls) or outcomes (not limited to SSHTBs) focused on a wider age range, or were published several years ago. A comprehensive search aimed to capture all the published literature on the topic. However, it is possible that some studies were inadvertently excluded, and publication bias may be an issue as the grey literature was not searched. Studies looking at exposure to suicides in the media were excluded so as to focus on the impact of close bereavements. Nonetheless, research shows that suicide clusters in young people may result from social transmission following media-reported suicides [66], which could not be captured in this review. Furthermore, research on the death of personally

known individuals may not necessarily constitute close relationships (e.g. estranged biological parents identified from national record studies). Recent research also highlights distinct processes in the development of suicidal thoughts and behaviours [e.g. 25–27]; in this review, we included studies that captured a range of outcomes, which may account for some of the inconsistent findings. Due to constraints on time and resources, only a portion of the data was double-extracted and assessed for risk of bias. However, both data extraction and risk of bias tools were pre-piloted and no major disagreements were raised.

The NOS used for assessing risk of bias within studies has been used widely in systematic reviews of non-randomised studies [70,71]. It has been deemed accessible, valid and reliable, and was adapted for the current review as per guidelines [31,72]. However, its reliability has been questioned more recently [73], in particular due to its low inter-rater reliability [74,75]. Despite this, kappa was high in this study. While the cross-sectional risk of bias tool used in this review has been employed in several studies [e.g. 32,76–78], its psychometric properties have yet to be established. Furthermore, depression was identified as a critical factor for which studies could be given an additional star rating when assessing bias; however, other pertinent factors such as prior history of SSHTBs may have been equally important to control for, given their strong association with future behaviours [53]. Regardless, several studies failed to control for either depression or self-harm history, and some did not account for potential confounding at all, so this decision likely had little impact on ratings. Finally, the tendency for newer studies to score lower on risk of bias may reflect the increasing trend of using such instruments in more recent times.

With regards to limitations stemming from the literature, the majority of studies were conducted in northern Europe and the United States. It is therefore unclear whether findings are generalisable to non-Western cultures, given that attitudes toward suicide may differ by country or culture [79]. Several register-based studies stemmed from Scandinavia, where populations tend to be predominantly white, have a relatively high socioeconomic status, and access to free universal health care [44]. Publications also suffered from potential selection bias due to convenience sampling, such as recruiting from higher education settings, obituaries, or newspaper advertisements. While many publications (mostly register-based) scored low on risk of bias, others failed to present pertinent information including details about the methodology or analyses conducted, or even age distributions. This raises questions about the quality and representativeness of some data. Furthermore, while our inclusion criteria was aimed at capturing the experience of individuals between 12 to 18 years old, this also meant that some younger and older participants were included, given that most studies employed broad age criteria. Acknowledging that definitions of adolescence vary, and "young people" may be defined as individuals up to 24 years of age [80], we chose not to impose an upper age limit for the included studies, providing that they met the other criteria for inclusion. After consideration, we also excluded publications which focused solely on the experience of 18+ year olds, given that they tended to report an older average age and therefore would not truly reflect the experience of adolescent groups, even if they included some 18 year old participants. Nonetheless, given the variability in inclusion criteria across individual studies, it is still possible that those publications ultimately included do not necessarily capture the experience of all bereaved adolescents. There was also an absence of qualitative studies which met the inclusion criteria.

## Implications and summary

Despite some contradictory evidence regarding the relative risk of SSHTBs, the studies included in this review reinforce the view that suicide bereaved adolescents are at risk of similar outcomes themselves. As consequences can be long term, even spanning several decades,

therapeutic and preventive measures should therefore enquire about past experiences of loss. Those working with young people should particularly probe beyond the experience of immediate family losses, as the few existing studies on non-familial bereavements to date suggest that the death of a friend may be influential among adolescents. School-based interventions may be well placed to identify individuals at risk, given their role as a primary source of contact for young people. There is some evidence of the effectiveness of school-based programmes, e.g. in improving knowledge of suicide, attitudes toward suicide, and help-seeking behaviours for oneself or a peer [81]. However, there is limited research available on postvention programmes within schools, and further research on this would be beneficial; e.g. examining their effectiveness in reducing suicide clusters. Additional studies are needed to understand how adolescents who do not present to clinical settings are affected by bereavement, as findings from self-report studies were inconclusive. Research shows that self-harm is prevalent but more hidden in the community [69], so studies of community adolescents using validated measures and larger representative samples are essential. Work based outside of Europe and the United States is also called for given the lack of research in these areas.

Although exposure to suicide was associated with increased risk of some SSHTB outcomes, it is worth noting that other bereavements also led to similar outcomes for young people. Research and strategies addressing suicide should therefore look beyond mere cause of death and focus on the broader context of loss. For instance, research with young adults (18–40 years old) has highlighted stigma as a potential moderator of the relationship between bereavement and suicide attempts [82–84], and other variables such as family environment [85], closeness of the relationship or quality of remaining social supports may be particularly influential [20]. Theory-based research can help understand the broader context of bereavement, as it is acknowledged that suicide results from an interplay of biopsychosocial factors. Increasing recognition has been given to the role of suicide bereavement in such frameworks; e.g. the Integrated Motivational-Volitional Model of Suicide views exposure to the self-harm of others as a key factor in the transition from ideation to attempts [26,86], but research testing this assumption with adolescent populations is scarce. As the findings of this review suggest, factors which predict ideation may differ from those which predict behavioural enaction, which is an important distinction for research going forward. Overall, research, policy, and programmes to reduce suicide should consider the impact of suicide bereavement, and acknowledge the wider context and circumstances of the individual, such as age, development, and social context, so that effective postvention can benefit those most in need.

## Supporting information

**S1 File. PROSPERO protocol.**
(PDF)

**S2 File. PRISMA checklist.**
(DOC)

## Acknowledgments

We would like to express our gratitude to Bethany Martin, who contributed to the screening of records, and Amanda Rhona Brown, who contributed to piloting of the data extraction form.

## Author Contributions

**Conceptualization:** Laura del Carpio, Sally Paul, Susan Rasmussen.

**Data curation:** Laura del Carpio.

**Formal analysis:** Laura del Carpio, Sally Paul, Abigail Paterson, Susan Rasmussen.

**Funding acquisition:** Laura del Carpio, Sally Paul, Susan Rasmussen.

**Investigation:** Laura del Carpio, Abigail Paterson.

**Methodology:** Laura del Carpio, Sally Paul, Susan Rasmussen.

**Project administration:** Laura del Carpio, Sally Paul, Susan Rasmussen.

**Resources:** Laura del Carpio.

**Supervision:** Sally Paul, Susan Rasmussen.

**Validation:** Laura del Carpio, Abigail Paterson.

**Visualization:** Laura del Carpio, Sally Paul, Susan Rasmussen.

**Writing – original draft:** Laura del Carpio.

**Writing – review & editing:** Laura del Carpio, Sally Paul, Abigail Paterson, Susan Rasmussen.

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
