## [Decision Letter · Decision Letter 0]

9 Dec 2020

PONE-D-20-33234

A systematic review of controlled studies of suicidal and self-harming behaviours in adolescents following bereavement by suicide

PLOS ONE

Dear Dr. Rasmussen,

Thank you for submitting your manuscript to PLOS ONE. After careful consideration, we feel that it has merit but does not fully meet PLOS ONE’s publication criteria as it currently stands. Therefore, we invite you to submit a revised version of the manuscript that addresses the points raised during the review process.

We look forward to receiving your revised manuscript.

Kind regards,

Michelle Tye, Ph.D.

Academic Editor

PLOS ONE

Journal Requirements:

2. Our internal editors have looked over your manuscript and determined that it may be within the scope of our Understanding and Preventing Suicide Call for Papers. This collection of papers is headed by a team of Guest Editors for PLOS ONE: Jo Robinson, Merike Sisask, Kairi Kõlves and Maria Oquendo. With this collection we hope to bring together researchers exploring  suicide and self-harm from societal, psychological, spiritual, clinical, and biological perspectives. Additional information can be found on our announcement page: https://collections.plos.org/s/understanding-suicide. If you would like your manuscript to be considered for this collection, please let us know in your cover letter and we will ensure that your paper is treated as if you were responding to this call. Agreeing to be part of the call-for-papers will not affect the date your manuscript is published. If you would prefer to remove your manuscript from collection consideration, please specify this in the cover letter.

Additional Editor Comments (if provided):

While there is general agreement regarding the importance of this topic - adolescents bereaved by suicide - and implications for national suicide prevention targets for the future, there are some considerable methodological limitations, particularly in respect to the ambitious aims of the paper. I strongly agree with the major points raised by each reviewer, and appropriate consideration of them will markedly strengthen the manuscript.

Reviewers' comments:

Reviewer's Responses to Questions

**Comments to the Author**

1. Is the manuscript technically sound, and do the data support the conclusions?

Reviewer #1: Partly

Reviewer #2: Partly

Reviewer #3: Partly

2. Has the statistical analysis been performed appropriately and rigorously? 

Reviewer #1: N/A

Reviewer #2: N/A

Reviewer #3: N/A

3. Have the authors made all data underlying the findings in their manuscript fully available?

Reviewer #1: Yes

Reviewer #2: Yes

Reviewer #3: Yes

4. Is the manuscript presented in an intelligible fashion and written in standard English?

Reviewer #1: Yes

Reviewer #2: No

Reviewer #3: Yes

5. Review Comments to the Author

Reviewer #1: This review has been submitted on a topic where there is a growing number of systematic reviews, including one meta-analysis are currently available. Therefore when reviewing this manuscript I focussed largely on the quality of evidence presented and what it contributes to the field. The findings are consistent with those published in a recent influential meta-analysis in the field. A stronger rationale is required by the authors to justify how the present study advanced exisiting findings. My main feedback is provided below:

1. The number of annual suicides reported globally needs to be revised (Line 50- 8 million is grossly overestimated).

2. Recent meta-analytic evidence that examined the association between exposure to suicide and suicide attempt on subsequent suicide-related outcomes which found limited evidence to suggest exposure to suicide had different effects for young people versus adults (see doi:10.1371/journal.pmed.1003074). The findings from this systematic review and meta-analysis does not seem to be considered in the introduction, which is noteworthy given the overlap in the content. This is also true of the conclusions drawn by the authors.

3. The decision to include only those bereaved by other causes needs to be carefully considered in terms of the evidence and conclusions that can be drawn. From a study design perspective, comparing those who are bereaved by other causes is problematic because observed differences in the exposed group are likely to be smaller, or non-existent. This means that the unique experiences of suicide bereaved people may be significantly underestimated (see doi.org/10.1017/S2045796019000581).

4. The authors state their objective “.. this review was to explore the evidence on whether bereavement by suicide confers greater risk of self-harm or suicidal outcomes (thoughts and behaviours)”. However recent evidence suggests that combiding thoughts and behaviours in the context of suicide bereavement results in inconsistent results. This limitation warrants acknowledgement by the authors.

5. Although the authors have noted some overlap between populations in table one a number of studies are missing superscripts indicating the overlap between populations (ie. Niederkrotenthaler and Li)

6. Authors did not indicate whether the exposure to bereavement occurred before or after the outcome of interest (see doi: 10.1371/journal.pmed.1003074). This means that it cannot be ruled out that the outcomes measured by some studies included in the review occurred before the exposure of interest (ie. suicide or suicide attempt). Ideally Table 1 should indicate whether the study controlled for whether the exposure occurred before the outcome and the potential biases associated with those studies that do not control for this.

1. “A narrative synthesis of studies was planned, given the substantial heterogeneity in research designs and methodologies across studies.” I disagree with this statement. I understand that this is commonly used but there is growing recognition in the field of systematic reviews and research methodology that heterogeneity is not alone sufficient for narrative synthesis (especially since the authors have not demonstrated/quantified heterogeneity to support this statement – i.e., through a Tau2 of I2 statistic see doi: 10.1136/bmj.a117).

2. The authors have not acknowledged deviations from the protocol registered in Prospero (e.g., inclusion of qualitative studies and the additional outcomes reported).

7. “Pitman et al. [16] evaluated studies of bereavement by suicide versus other types of death. With regards to suicide risk, strong evidence of increased risk was found among adults bereaved by the suicide of a partner or ex-partner, and mothers bereaved of an adult child, compared to other causes of death.” (Line 76-78) This statement needs to be tempered considered the confidence intervals reported by Pitman et al which in fact suggest that there was significant overlap between suicide risk in those bereaved by suicide versus bereavement from other causes of death.

8. “Focusing on control group studies is necessary to clarify whether suicide bereavement specifically confers greater risk of SSHTBs compared” (Line 105). Im not entirely convinced that cross sectional studies have a control group. I believe it is more accurate to refer to these as a comparison group.

Minor feedback;

9. “A dose-response relationship was found..” (Line 305). With only two studies it is preferable to refer to this as a potential dose response relationship.

10. “In addition to the two papers discussed in 3.5.4.1…” (Line 386) this is a typo.

11. It is preferable to use different terminology ‘decendent’ is a largely legal term that has been identified by those with lived experience to be stigmatic (Line 541).

12. It is preferable that explanatory mechanisms (Line 635) be replaced with possible mechanisms

Reviewer #2: Although I generally admire articles like this, synthesizing large numbers of studies on a particular theme, ferreting out their commonalities and divergences, and then offering more empirically guided viewpoints on the needs and the necessary directions for future research, this study on adolescent suicidal self-harm and bereavement seemed to focus on too many elements at the same time to be readily digestible. Had the authors focused more exclusively on the question of whether adolescent suicide is higher when a parent dies by suicide, rather than from another diverging cause, that in my mind, would have been a sufficiently important enough focus for this paper. Adding all the extra elements of suicidal self-harm and the circumstances surrounding the deaths seemed to overload this paper. By dividing this paper into two synthetic summaries there would also be less vagueness and imprecision about why particular studies were included or excluded from the comparisons. Simplifying the focus of this paper would go a long way toward making this a clearer and more coherent piece of synthesizing research.

Reviewer #3: Thank you for allowing me to review this manuscript aiming to systematically summarize the literature on the impact of suicide on adolescents, and specifically the risk of suicidal behaviour following bereavement. The topic of this paper is important both from a clinical and a public health perspective. Findings may indicate a need to provide adequate support to adolescents bereaved by suicide. The study may also underscore that young people bereaved by suicide should be considered a potential target group in national suicide prevention policies.

The research questions were clearly formulated, but they are also very ambitious. For example, answering research question 1 ideally would involve a meta-analysis of pooled data. Answering research question 3 could have involved an evaluation of the psychometric characteristics of the instruments used in studies. Apparently, the latter was beyond the scope of this review. The answer to research question 3 seems to be limited to a discussion of research methods. There is nothing wrong with that, but I wonder if research question 3 should be reformulated.

While I acknowledge the importance of the topic and the hard work that authors have put into this paper there are concerns that need attention.

A crucial concern is that the selected studies do not seem to match with the age range criterium that was stipulated in the inclusion criteria (lines 136-140). According to the inclusion criteria, selected studies “needed to include adolescent participants, who were defined as being between the ages of 12-18 years old at the time of assessment of outcomes”. However, several of the selected studies explicitly excluded that age group.

For example, the study by Tal et al. (2017) included participants aged 18-95 years with a mean of 47 years (participants bereaved by suicide). Years since the suicide was M =3.9 years, SD = 4.6. Also, the introduction and discussion of this paper did not mention anything about adolescents.

Other example: the follow-up study by Kolves et al. (2020) included participants aged 18+ with a mean age of 52.70 for those bereaved by suicide. Potential participants were approached to take part in the study about 5 months after the loss.

Other example: Rostila et al. (2012). On page 339 it says: “Individuals were stratified into 5 different groups consisting of people who experienced sibling loss at ages 18–29 years, 30–39 years, 40–49 years, 50–59 years, and 60–69 years”.

I haven’t checked all 30 studies included in this review, but it is obvious that several studies have not investigated adolescents, aged 12-18 years.

Further, the review also included studies (such as Pitman) with participants 18+, in which participants may have experienced the death before they were aged 18. However, according to the inclusion criteria, these studies should have been excluded as well.

In the same vein: Line 221: ‘Nine studies recruited individuals aged 18 or above’.

Lines 695-697: Strengths and limitations: It does not make sense.

Moreover, how come that the study of these authors was not included?

Del Carpio, L., Rasmussen, S., & Paul, S. (2020). A theory-based longitudinal investigation examining predictors of self-harm in adolescents with and without bereavement experiences. Frontiers in Psychology, 11. https://pubmed.ncbi.nlm.nih.gov/32581958/

I do not understand how such a mismatch could have occurred and I strongly recommend that authors reconsider the manuscript.

Given the above, I hesitate to provide comments on the rest of the manuscript. Still, please find a few comments or questions below.

Abstract, line 41: elevated risk of what?

Conclusions of abstract: Not clear how these follow from the review.

Introduction, line 50

According to the WHO, suicide accounts for approx. 800,000 deaths per year.

Not sure what authors mean with ‘8 million’.

Study selection, lines 165-166

Please specify how many ‘cases’ were referred to the second and third reviewer,

Data extraction, lines 168-174

Data extraction is usually done by two researchers to ensure reliability. Please motivate why this was done by one researcher only?

A second researcher extracted data from one third of the papers. How much disagreement was there? How do you know if data extraction of the remaining twenty papers was correct?

Risk of bias, line 175

The same questions apply here. Assessment of quality or risk of bias of studies is usually conducted by two researchers. Please motivate why this was done by one researcher only?

How do you know if the assessment of the remaining twenty papers was correct?

Planned methods of analysis, lines 189-196

Not sure what you mean with ‘timing or age groups’.

I was wondering if authors could specify the planned methods separately for each research question. It is not clear if the currently presented planned methods of analysis address the third research question.

Line 207: Prisma figure

The box ‘qualitative synthesis’ might be better ‘narrative synthesis’.

Line 241: Risk of bias: Overall, it seems that more recent studies have a better score than older studies (though not always), which have been conducted before the quality instruments became more is use.

Line 271: Association between bereavement and subsequent suicide

The description of these studies (population-based national registers, case-control, cohort) makes me wonder if a meta-analysis for these studies would be possible. Looking at the summary description in this section, it seems that all these studies provided data that can be pooled. It would certainly add value to the review.

I am also wondering if the data on attempted suicide as an outcome would be better placed in one of the next sections, which are about (hospitalized) attempted suicide and self-reported suicidal behaviour.

Line 326: Which confounding variables?

Line 331: ‘However, the differences were not statistically significant’.

Do you mean the differences between boys and girls? Please specify.

Line 337: What is ‘intentional self-harm mortality’? Maybe it is easier to talk about suicide. If not, please provide a definition.

Line 355: Data regarding attempted suicide as an outcome might be better placed in one of the next sections dedicated to attempted suicide as an outcome.

Line 384: Also in this section I am wondering if it would be possible to pool some of the data. It would give so much more power to the review.

Line 386: It is not clear what ‘discussed in 3.5.4.1’ means.

Line 434: the subheading ’relationship’ is missing in this section.

Line 635: In this section on ‘Explanatory mechanisms’ it seems that suicide, attempted suicide and suicidal ideation are mingled. Would it be possible to be more clear what kind of mechanisms would be applicable for the different types of suicidal behaviour?

Line 665: A mentioned above, it is an interesting part of the manuscript, but it more about the data sources and study methods than about measures or instruments.

Line 744: What is the reason for suddenly referring to ‘suicide clusters’? The occurrence of clusters is often influenced by exposure to media or social media messages about suicide or suicidal behaviour. I would hesitate to introduce concepts that have not been addressed by the review.

To conclude this review, given the importance of the topic and the shortage of research in this field I hope that this paper will get through the review process. Hence, I hope that these few comments may help improving the manuscript, and I wish the authors all the best with the revision!

6. PLOS authors have the option to publish the peer review history of their article (what does this mean?). If published, this will include your full peer review and any attached files.

Reviewer #1: No

Reviewer #2: No

Reviewer #3: No

---

## [Author Response · Author response to Decision Letter 0]

25 Jan 2021

Journal Requirements: 

Response: We have revised the figure according to the guidelines provided. Please let us know if there are any additional changes required to the formatting. 

2. Our internal editors have looked over your manuscript and determined that it may be within the scope of our Understanding and Preventing Suicide Call for Papers. This collection of papers is headed by a team of Guest Editors for PLOS ONE: Jo Robinson, Merike Sisask, Kairi Kõlves and Maria Oquendo. With this collection we hope to bring together researchers exploring suicide and self-harm from societal, psychological, spiritual, clinical, and biological perspectives. Additional information can be found on our announcement page: https://collections.plos.org/s/understanding-suicide. If you would like your manuscript to be considered for this collection, please let us know in your cover letter and we will ensure that your paper is treated as if you were responding to this call. Agreeing to be part of the call-for-papers will not affect the date your manuscript is published. If you would prefer to remove your manuscript from collection consideration, please specify this in the cover letter.

Response: Thank you for bringing this to our attention. We would appreciate being considered for this Call for Papers. 

Reviewer #1: 

Reviewer #1: This review has been submitted on a topic where there is a growing number of systematic reviews, including one meta-analysis are currently available. Therefore when reviewing this manuscript I focussed largely on the quality of evidence presented and what it contributes to the field. The findings are consistent with those published in a recent influential meta-analysis in the field. A stronger rationale is required by the authors to justify how the present study advanced exisiting findings. My main feedback is provided below:

1. The number of annual suicides reported globally needs to be revised (Line 50- 8 million is grossly overestimated).

Response: We apologise for this typographical error which went unnoticed. We have now corrected this. 

2. Recent meta-analytic evidence that examined the association between exposure to suicide and suicide attempt on subsequent suicide-related outcomes which found limited evidence to suggest exposure to suicide had different effects for young people versus adults (see doi:10.1371/journal.pmed.1003074). The findings from this systematic review and meta-analysis does not seem to be considered in the introduction, which is noteworthy given the overlap in the content. This is also true of the conclusions drawn by the authors.

Response: Thank you for pointing this out, this was an unintentional omission which should have been included. We have updated the introduction and discussion sections to incorporate this study: 

More recently, Hill et al. [22] reviewed studies on exposure to suicide and suicide attempts, and found that exposure to suicide was associated with an increased odds of suicide and suicide attempts, though not suicidal ideation. Age was not found to moderate risk, as shown through comparisons of studies with a majority of youth (aged 25 or under) compared to adult participants. However, this meta-analysis excluded studies where control groups were comprised of participants who had been exposed to other modes of death. 

These findings are largely consistent with conclusions by Hill et al. [22] in their recent review, which indicated an elevated risk of suicide and suicide attempts but not suicidal ideation among those exposed to suicide.

3. The decision to include only those bereaved by other causes needs to be carefully considered in terms of the evidence and conclusions that can be drawn. From a study design perspective, comparing those who are bereaved by other causes is problematic because observed differences in the exposed group are likely to be smaller, or non-existent. This means that the unique experiences of suicide bereaved people may be significantly underestimated (see doi.org/10.1017/S2045796019000581).

Response: Thank you for raising this point. We acknowledge that there are limitations to using case-control studies, particularly where group differences are small. However, we were interested specifically in whether there are unique aspects to the experience of suicide loss compared to other deaths. Using non-bereaved controls would not have allowed us to draw conclusions about whether outcomes observed were specific to experiencing a suicide, rather than any death generally. 

We also recognise that including all non-suicide bereaved groups as controls may impact our findings by combining what may be very heterogeneous experiences, e.g. there may be fewer differences between those affected by suicide and other sudden or traumatic deaths (16; 18). Similarly, subgroup differences may exist within those bereaved by suicide. For this reason, we have considered cause of death when examining bereavement characteristics, but given the lack of previous reviews in this area with a focus on adolescents, we decided to include all non-suicide bereaved groups as comparators. 

We have added a sentence in the Objectives to expand on this:

Studies using solely non-bereaved controls limit conclusions that can be drawn about whether the experiences of a suicide-bereaved group are unique to this type of loss rather than bereavement experiences generally. 

4. The authors state their objective “.. this review was to explore the evidence on whether bereavement by suicide confers greater risk of self-harm or suicidal outcomes (thoughts and behaviours)”. However recent evidence suggests that combiding thoughts and behaviours in the context of suicide bereavement results in inconsistent results. This limitation warrants acknowledgement by the authors.

Response: We acknowledge that research in this area distinguishes between thoughts and behaviours related to suicide. We aimed to capture this to some extent in the structuring of our results (e.g., considering suicide deaths as a separate outcome), but have added a sentence in the discussion to acknowledge this limitation: 

Recent research also highlights distinct processes in the development of suicidal thoughts and behaviours [e.g. 25, 26, 27]; in this review, we included studies that captured a range of outcomes, which may account for some of the inconsistent findings.

5. Although the authors have noted some overlap between populations in table one a number of studies are missing superscripts indicating the overlap between populations (ie. Niederkrotenthaler and Li)

Response: Thank you for this comment. We checked through the papers, and where we have found overlap, we have indicated this. This includes the addition of a superscript for Kuramoto et al., 2010 and 2013 studies, as well as Burrell et al., 2017 and 2018. 

The reviewer has specifically highlighted the studies by Niederkrotenthaler et al. and Li et al.; however, from our examination of these papers, we could not see where the overlap was, and have therefore not highlighted it. We would be happy for more clarification from the reviewer. 

6. Authors did not indicate whether the exposure to bereavement occurred before or after the outcome of interest (see doi: 10.1371/journal.pmed.1003074). This means that it cannot be ruled out that the outcomes measured by some studies included in the review occurred before the exposure of interest (ie. suicide or suicide attempt). Ideally Table 1 should indicate whether the study controlled for whether the exposure occurred before the outcome and the potential biases associated with those studies that do not control for this.

Response: All studies looked at outcomes following bereavement (some may also have included a measure of pre-bereavement or lifetime engagement in suicidal thoughts or behaviours). We included information in the table about which measures were used in each study to capture these outcomes, and an indication of timing is given where relevant; e.g., “Beck Scale for Suicide Ideation measured presence and intensity of suicidal intent in the previous week”. 

1. “A narrative synthesis of studies was planned, given the substantial heterogeneity in research designs and methodologies across studies.” I disagree with this statement. I understand that this is commonly used but there is growing recognition in the field of systematic reviews and research methodology that heterogeneity is not alone sufficient for narrative synthesis (especially since the authors have not demonstrated/quantified heterogeneity to support this statement – i.e., through a Tau2 of I2 statistic see doi: 10.1136/bmj.a117).

Response: Thank you for pointing this out, we agree that the rationale for our decision should have been made stronger here. We stated that substantial heterogeneity precluded meta-analysis; in this case, heterogeneity was not limited to statistical origin, but also to clinical and methodological differences between studies. Specifically, there was large variability in participants, exposures, and outcomes measured, as well as study designs and quality, which may have resulted in inappropriate and less meaningful summaries of findings through the use of meta-analysis (Deeks et al., 2020; Haidich, 2010). We have rephrased this section to be more specific on why we have chosen to perform a narrative synthesis: 

A narrative synthesis of studies was planned, given the substantial heterogeneity in sample populations, exposures and outcomes measured, research designs and quality, and statistical methods across studies, which deemed such an analysis more appropriate. 

2. The authors have not acknowledged deviations from the protocol registered in Prospero (e.g., inclusion of qualitative studies and the additional outcomes reported).

Response: No qualitative studies which met inclusion criteria were identified, as noted in the discussion under strengths and limitations. A statement to this effect has now also been added to the Results under the section “Study characteristics”. 

We considered variables which may be related to risk of self-harm/suicide under the sections of ‘bereavement characteristics’, ‘characteristics of the deceased person’, and ‘participant characteristics’. We acknowledge that there is room to explore a multitude of additional variables, but we chose to limit our discussion to those relevant to the experience of loss, as noted in the paragraph under “Synthesis of results”. 

7. “Pitman et al. [16] evaluated studies of bereavement by suicide versus other types of death. With regards to suicide risk, strong evidence of increased risk was found among adults bereaved by the suicide of a partner or ex-partner, and mothers bereaved of an adult child, compared to other causes of death.” (Line 76-78) This statement needs to be tempered considered the confidence intervals reported by Pitman et al which in fact suggest that there was significant overlap between suicide risk in those bereaved by suicide versus bereavement from other causes of death.

Response: Thank you for pointing this out, we have removed the word “strong” from this sentence. 

8. “Focusing on control group studies is necessary to clarify whether suicide bereavement specifically confers greater risk of SSHTBs compared” (Line 105). Im not entirely convinced that cross sectional studies have a control group. I believe it is more accurate to refer to these as a comparison group.

Response: We chose to use “control group” on the basis of previous literature, but acknowledge the reviewer’s comments, and have amended the wording of the “Eligibility criteria” to read, “…comparing exposure to suicide deaths to at least one other non-suicide bereaved comparison or control group”. 

Minor feedback;

9. “A dose-response relationship was found..” (Line 305). With only two studies it is preferable to refer to this as a potential dose response relationship.

Response: We have changed this statement to read “a potential dose-response relationship”. 

10. “In addition to the two papers discussed in 3.5.4.1…” (Line 386) this is a typo.

Response: Apologies, this was a typographical error which has now been corrected to “…the two papers discussed above”. 

11. It is preferable to use different terminology ‘decendent’ is a largely legal term that has been identified by those with lived experience to be stigmatic (Line 541).

Response: Thank you for noting this, we have changed this wording throughout, i.e. replaced with “the deceased person” or “the person who died”.

12. It is preferable that explanatory mechanisms (Line 635) be replaced with possible mechanisms

Response: We have replaced “explanatory mechanisms” with “possible mechanisms”.

Reviewer #2:

Reviewer #2: Although I generally admire articles like this, synthesizing large numbers of studies on a particular theme, ferreting out their commonalities and divergences, and then offering more empirically guided viewpoints on the needs and the necessary directions for future research, this study on adolescent suicidal self-harm and bereavement seemed to focus on too many elements at the same time to be readily digestible. Had the authors focused more exclusively on the question of whether adolescent suicide is higher when a parent dies by suicide, rather than from another diverging cause, that in my mind, would have been a sufficiently important enough focus for this paper. Adding all the extra elements of suicidal self-harm and the circumstances surrounding the deaths seemed to overload this paper. By dividing this paper into two synthetic summaries there would also be less vagueness and imprecision about why particular studies were included or excluded from the comparisons. Simplifying the focus of this paper would go a long way toward making this a clearer and more coherent piece of synthesizing research.

Response: Thank you for the feedback, we appreciate the comments made by the reviewer. We acknowledge that this review presents a vast amount of information to readers, and this was an issue we considered in our preparation of the manuscript. While we feel there is value in examining how parental suicide specifically impacts adolescents, we chose to examine the issue of whether suicide bereavement differs to other modes of death for a number of reasons. 

There has been ongoing debate in the literature on whether suicide poses unique challenges to those bereaved compared to those who have lost a loved one to other deaths, and if this is the case, would strengthen the argument for targeted interventions for people who have experienced such losses. The development of effective postvention is still underway, and can be strengthened by findings such as these, which might implicate specific factors to target. Our understanding of suicide from a theoretical standpoint also benefits from exploring how different losses, as well as circumstances surrounding the death, impact the wider context of an individual, and points to opportunities for suicide prevention. Finally, research has advanced since the earlier reviews conducted in this area, and other authors have focused on older age groups or taken a different focus in terms of outcomes, etc. We therefore felt the focus of this study fills a necessary and specific gap in our knowledge about adolescent experiences of loss by suicide compared to other deaths, which we felt was important to address as a single synthesis. 

Reviewer #3:

Reviewer #3: Thank you for allowing me to review this manuscript aiming to systematically summarize the literature on the impact of suicide on adolescents, and specifically the risk of suicidal behaviour following bereavement. The topic of this paper is important both from a clinical and a public health perspective. Findings may indicate a need to provide adequate support to adolescents bereaved by suicide. The study may also underscore that young people bereaved by suicide should be considered a potential target group in national suicide prevention policies.

The research questions were clearly formulated, but they are also very ambitious. For example, answering research question 1 ideally would involve a meta-analysis of pooled data. Answering research question 3 could have involved an evaluation of the psychometric characteristics of the instruments used in studies. Apparently, the latter was beyond the scope of this review. The answer to research question 3 seems to be limited to a discussion of research methods. There is nothing wrong with that, but I wonder if research question 3 should be reformulated.

Response: Thank you for these comments. We agree that the rationale for our decision to not conduct a meta-analysis should have been made stronger. We stated that substantial heterogeneity precluded meta-analysis; in this case, heterogeneity was not limited to statistical origin, but also to clinical and methodological differences between studies. Specifically, there was large variability in participants, exposures, and outcomes measured, as well as study designs and quality, which may have resulted in inappropriate and less meaningful summaries of findings through the use of meta-analysis (Deeks et al., 2020; Haidich, 2010). We have rephrased this section to be more specific on why we have chosen to perform a narrative synthesis: 

A narrative synthesis of studies was planned, given the substantial heterogeneity in sample populations, exposures and outcomes measured, research designs and quality, and statistical methods across studies, which deemed such an analysis more appropriate. 

We also recognise the value of approaching research question 3 as an evaluation of the psychometric properties of the instruments used, and there could be value in doing this going forward. However, in order not to deviate from the protocol, we chose to limit our discussion to the methods employed by the studies. We agree with the reviewer that the research question could be reformulated, and we have done this in a way to reflect the original wording of the published protocol: “Finally, which measures have been used to capture outcomes in the literature?”

While I acknowledge the importance of the topic and the hard work that authors have put into this paper there are concerns that need attention.

A crucial concern is that the selected studies do not seem to match with the age range criterium that was stipulated in the inclusion criteria (lines 136-140). According to the inclusion criteria, selected studies “needed to include adolescent participants, who were defined as being between the ages of 12-18 years old at the time of assessment of outcomes”. However, several of the selected studies explicitly excluded that age group.

For example, the study by Tal et al. (2017) included participants aged 18-95 years with a mean of 47 years (participants bereaved by suicide). Years since the suicide was M =3.9 years, SD = 4.6. Also, the introduction and discussion of this paper did not mention anything about adolescents.

Other example: the follow-up study by Kolves et al. (2020) included participants aged 18+ with a mean age of 52.70 for those bereaved by suicide. Potential participants were approached to take part in the study about 5 months after the loss.

Other example: Rostila et al. (2012). On page 339 it says: “Individuals were stratified into 5 different groups consisting of people who experienced sibling loss at ages 18–29 years, 30–39 years, 40–49 years, 50–59 years, and 60–69 years”.

I haven’t checked all 30 studies included in this review, but it is obvious that several studies have not investigated adolescents, aged 12-18 years.

Further, the review also included studies (such as Pitman) with participants 18+, in which participants may have experienced the death before they were aged 18. However, according to the inclusion criteria, these studies should have been excluded as well.

In the same vein: Line 221: ‘Nine studies recruited individuals aged 18 or above’.

Response: All studies chosen for inclusion needed to include at least some individuals within the stipulated 12-18 year old bracket, as per our inclusion criteria. We understand the reviewer’s concerns, and recognise that this may result in research with an older average age range being included. Very few studies included solely individuals within our age group of interest, as most included younger, or more commonly, older participants. Therefore, sourcing data exclusively on the 12-18 year old age range across all studies was not feasible and problematic. Given that definitions of adolescence vary, and the WHO refers to “young people” as those between 10 and 24 years of age [90], we therefore decided not to impose an upper limit on participant age. We chose instead to be consistent with our decisions by including all studies with at least some individuals falling into our desired age bracket. We did, however, reject studies focusing solely on parents whose offspring had died by suicide, as they were unlikely to represent our population of interest.

We do appreciate the reviewer’s concerns on this, and have added further discussion of this under the limitations. This includes the following: 

Furthermore, while our inclusion criteria meant that we chose to include studies involving at least some individuals aged between 12 to 18 years old, this also resulted in a number of studies which perhaps represented the experiences of an older age range than we sought. Acknowledging that definitions of adolescence vary, and “young people” may be defined as individuals up to 24 years of age [90], we chose not to impose an upper age limit for the included studies, providing that they met the minimum criteria for inclusion. Nonetheless, it is possible that not all studies included here may fully capture the experience of bereaved adolescents.

Lines 695-697: Strengths and limitations: It does not make sense.

Response: We have reworded the first two sentences of this section for clarity. 

Moreover, how come that the study of these authors was not included?

Del Carpio, L., Rasmussen, S., & Paul, S. (2020). A theory-based longitudinal investigation examining predictors of self-harm in adolescents with and without bereavement experiences. Frontiers in Psychology, 11. https://pubmed.ncbi.nlm.nih.gov/32581958/

Response: Thank you for pointing this out. We were aware of this paper, however as we conducted the last update to the literature search in May 2020, this paper was not available for inclusion. 

I do not understand how such a mismatch could have occurred and I strongly recommend that authors reconsider the manuscript.

Given the above, I hesitate to provide comments on the rest of the manuscript. Still, please find a few comments or questions below.

Abstract, line 41: elevated risk of what?

Response: We have added “of suicidal outcomes” to clarify the elevated risk. 

Conclusions of abstract: Not clear how these follow from the review.

Response: We have added a sentence summarising the review findings in this section for clarity: “Findings suggest that suicide loss is associated with subsequent suicide, and may be associated with non-fatal self harm.”

Introduction, line 50

According to the WHO, suicide accounts for approx. 800,000 deaths per year.

Not sure what authors mean with ‘8 million’.

Response: We apologise for this typographical error which went unnoticed. We have now corrected this. 

Study selection, lines 165-166

Please specify how many ‘cases’ were referred to the second and third reviewer,

Response: Twenty one difficult cases were referred to the second reviewer for discussion, and consensus was reached for all of these. 

Data extraction, lines 168-174

Data extraction is usually done by two researchers to ensure reliability. Please motivate why this was done by one researcher only?

A second researcher extracted data from one third of the papers. How much disagreement was there? How do you know if data extraction of the remaining twenty papers was correct?

Response: Time and resource limitations precluded a second independent review of the remaining 20 papers. Nevertheless, there can be confidence in the remaining papers as we followed the protocol (including pre-piloting data extraction forms and risk of bias tools), which yielded high kappa agreement (κ = .839) across all items with an independent reviewer on a third of the papers. The nature of the disagreements in the data extraction and risk of bias assessments were small points of detail that were not critical and were resolved by discussion. The papers which were chosen for review by the second researcher were selected randomly and from across the three study designs. Previous reviews have taken a similar approach to double extracting only a proportion of papers. We therefore felt that a sample of one third was appropriate. 

Risk of bias, line 175

The same questions apply here. Assessment of quality or risk of bias of studies is usually conducted by two researchers. Please motivate why this was done by one researcher only?

How do you know if the assessment of the remaining twenty papers was correct?

Response: As stated above, this decision was made due to constraints on time and resources, but given our adherence to the protocol and high kappa agreement resulting in only minor points of disagreement that were resolved by consensus, we are confident with the remaining papers. 

Planned methods of analysis, lines 189-196

Not sure what you mean with ‘timing or age groups’.

I was wondering if authors could specify the planned methods separately for each research question. It is not clear if the currently presented planned methods of analysis address the third research question.

Response: We have amended the phrase “timing or age groups” to read “timing of the death, and age groups”. 

We have also reworded this paragraph to address all research questions, and added a sentence to be more specific about the third research question: 

… To address RQ1 and 2, details of each study were reported under the broad domains of: bereavement circumstances, characteristics of the person who died, and characteristics of the bereaved individual. Discussion of the methods used to capture the outcomes in the literature (RQ3) are provided within the descriptions of each study. 

Line 207: Prisma figure

The box ‘qualitative synthesis’ might be better ‘narrative synthesis’.

Response: Thank you, we have updated this box. 

Line 241: Risk of bias: Overall, it seems that more recent studies have a better score than older studies (though not always), which have been conducted before the quality instruments became more is use.

Response: We agree this seems to be the case. We have added a comment on this in the discussion: “Finally, the tendency for newer studies to score lower on risk of bias may reflect the increasing trend of using such instruments in more recent times.”

Line 271: Association between bereavement and subsequent suicide

The description of these studies (population-based national registers, case-control, cohort) makes me wonder if a meta-analysis for these studies would be possible. Looking at the summary description in this section, it seems that all these studies provided data that can be pooled. It would certainly add value to the review.

Response: Thank you for your comments. As we have explained in our previous response above, we have amended the manuscript to provide a stronger rationale for not conducting a meta-analysis. 

I am also wondering if the data on attempted suicide as an outcome would be better placed in one of the next sections, which are about (hospitalized) attempted suicide and self-reported suicidal behaviour.

Response: We did include some commentary on hospitalisations for suicide attempt in this section as it was incorporated into the main study descriptions. As per the reviewer’s suggestion below, we have moved some of this information to the relevant next section on hospitalisations. 

Line 326: Which confounding variables?

Response: We have clarified the variables which were adjusted for: “after adjusting for age, sex, calendar time, and individual and family history of admission for mental illness”.

Line 331: ‘However, the differences were not statistically significant’.

Do you mean the differences between boys and girls? Please specify.

Response: We have amended the sentence to say: “However, the difference in risk after losing a father compared to a mother was not statistically significant (for both sons and daughters).” 

Line 337: What is ‘intentional self-harm mortality’? Maybe it is easier to talk about suicide. If not, please provide a definition.

Response: This study used ICD codes to capture mortality by type of death, in this case looking at deaths by suicide and intentional self-harm. We agree that this is a bit unclear and have reworded this phrase to avoid confusion. 

Line 355: Data regarding attempted suicide as an outcome might be better placed in one of the next sections dedicated to attempted suicide as an outcome.

Response: We had initially added this sentence under the section of suicide as an outcome in the interest of brevity, as the study reported on both the outcome of suicide death and psychiatric hospitalisations for suicide attempt. We agree with the reviewer and have moved this information to the following relevant section. 

Line 384: Also in this section I am wondering if it would be possible to pool some of the data. It would give so much more power to the review.

Response: Thank you for your comment. As we have explained in our previous response above, we have amended the manuscript to provide a stronger rationale for not conducting a meta-analysis. 

Line 386: It is not clear what ‘discussed in 3.5.4.1’ means.

Response: Apologies, this was a typographical error which has now been corrected to “…the two papers discussed above”. 

Line 434: the subheading ’relationship’ is missing in this section.

Response: All studies that measured hospitalisations for self-harm/suicide looked at the impact of parental deaths, therefore we chose not to include a separate section on relationships here, and instead commented on this at the start of the section. We included subsections only where there was relevant data to discuss, so consequently also looked at slightly different subcategories regarding participant characteristics across the three outcomes, guided by the available literature. 

Line 635: In this section on ‘Explanatory mechanisms’ it seems that suicide, attempted suicide and suicidal ideation are mingled. Would it be possible to be more clear what kind of mechanisms would be applicable for the different types of suicidal behaviour?

Response: We have added some clarification throughout this section on what types of thoughts or behaviours were being considered by the evidence presented. 

Line 665: A mentioned above, it is an interesting part of the manuscript, but it more about the data sources and study methods than about measures or instruments.

Response: As we noted in a previous response, we have reformulated Research Question 3 to align with the original protocol, which was to explore which measures were used within the literature to capture the outcomes of interest. 

Line 744: What is the reason for suddenly referring to ‘suicide clusters’? The occurrence of clusters is often influenced by exposure to media or social media messages about suicide or suicidal behaviour. I would hesitate to introduce concepts that have not been addressed by the review. 

Response: We considered the clustering of suicides within our section on “possible mechanisms” to explain some of the results of the review (e.g. familial clustering based on shared genetic vulnerabilities, social transmission through the media or knowing the deceased, etc.). We noted in the strengths and limitations that suicide clusters may result from social transmission following media-reported suicides, and we could not comment further on this as we excluded studies on exposure to suicide in the media (so as to focus on the impact of close bereavements). However, in the interest of discussing the implications of this work, we felt it important to acknowledge potential clusters within schools in the context of developing future postvention programmes. Supportive intervention for those bereaved and affected by suicide is recognised as key to responding to a suicide cluster [76]. For clarity, we have amended this sentence to read: 

“…however, there is limited research available on postvention programmes within schools, and further research on this would be beneficial; e.g. examining their effectiveness in reducing suicide clusters.”

To conclude this review, given the importance of the topic and the shortage of research in this field I hope that this paper will get through the review process. Hence, I hope that these few comments may help improving the manuscript, and I wish the authors all the best with the revision!

Response: We thank you for your helpful comments and suggestions. 

References: 

Haidich, A. B. (2010). Meta-analysis in medical research. Hippokratia, 14(Suppl 1): 29-37. 

Deeks JJ, Higgins JPT, Altman DG (editors). Chapter 10: Analysing data and undertaking meta-analyses. In: Higgins JPT, Thomas J, Chandler J, Cumpston M, Li T, Page MJ, Welch VA (editors). Cochrane Handbook for Systematic Reviews of Interventions version 6.1 (updated September 2020). Cochrane, 2020. Available from www.training.cochrane.org/handbook.

---

## [Decision Letter · Decision Letter 1]

4 May 2021

PONE-D-20-33234R1

A systematic review of controlled studies of suicidal and self-harming behaviours in adolescents following bereavement by suicide

PLOS ONE

Dear Dr. Rasmussen,

Thank you for submitting your manuscript to PLOS ONE. After careful consideration, we feel that it has merit but does not fully meet PLOS ONE’s publication criteria as it currently stands. Therefore, we invite you to submit a revised version of the manuscript that addresses the points raised during the review process.

I thank the authors for their efforts to address the reviewer comments, and note that some issues remain to be addressed - particularly those of Reviewer 3, which necessitates that the authors recheck their included study to ensure these align to the study criteria. 

We look forward to receiving your revised manuscript.

Kind regards,

Michelle Tye, Ph.D.

Academic Editor

PLOS ONE

Reviewers' comments:

Reviewer's Responses to Questions

**Comments to the Author**

1. If the authors have adequately addressed your comments raised in a previous round of review and you feel that this manuscript is now acceptable for publication, you may indicate that here to bypass the “Comments to the Author” section, enter your conflict of interest statement in the “Confidential to Editor” section, and submit your "Accept" recommendation.

Reviewer #1: (No Response)

Reviewer #2: All comments have been addressed

Reviewer #3: (No Response)

2. Is the manuscript technically sound, and do the data support the conclusions?

Reviewer #1: No

Reviewer #2: Yes

Reviewer #3: Partly

3. Has the statistical analysis been performed appropriately and rigorously? 

Reviewer #1: N/A

Reviewer #2: Yes

Reviewer #3: N/A

4. Have the authors made all data underlying the findings in their manuscript fully available?

Reviewer #1: No

Reviewer #2: Yes

Reviewer #3: Yes

5. Is the manuscript presented in an intelligible fashion and written in standard English?

Reviewer #1: Yes

Reviewer #2: Yes

Reviewer #3: Yes

6. Review Comments to the Author

Reviewer #1: I have considered the revisions made by the author and although it is clear that a lot of work went into this review the fundamental design limitations have not been properly addressed.

As I noted in my previous review that heterogeneity of this kind has and can be addressed through meta-analysis, in the very field that this review sits. It is important that we continue to move the field forward, building upon existing evidence. Not doing so, adds further noise to a field that has on more than one occasion suffered from spurious evidence. This is particularly the case when the authors have sought to answer what is essentially a quantitative question (Is suicidal behaviour unique to those who are bereaved by suicide vs other causes of death) with a narrative synthesis. The methods used in this review do not align with this type of research question and I stand by my previous comment that comparison to individuals bereaved by other causes of death do not inform whether the outcomes are unique to those bereaved by suicide. This research question alone has issues since most risk factors associated with suicidal behaviour have low specificity (i.e unique to suicidal behaviour in the first place.

Reviewer #2: Systematic review of adolescent suicide article REVISION

Although in my estimation this article submission has not been fully cleansed of all its minor and major imperfections, it is sufficiently on target to be of value to those attempting to make sense and coherent meaning from the large body of research out there on the effects of suicide bereavement upon adolescents, and will be of value to researchers and clinicians. The authors have worked diligently and effectively to address the criticisms offered by all reviewers and have responded satisfactorily to my own criticisms. I think there is one essential rewording required for the study abstract to make the study’s contents clearer. The first sentence of the methods section of the abstract should also include these words: “including several that extended beyond this age range.” With this amendment, I recommend acceptance of this manuscript for publication.

Reviewer #3: Dear authors thank you for your replies and for addressing my concerns. While I acknowledge that the manuscript has improved, I notice that one crucial concern has not been addressed. In my review I highlighted that several of the selected studies did not match with the age range criterium that was stipulated in the inclusion criteria. According to the inclusion criteria, selected studies “needed to include adolescent participants, who were defined as being between the ages of 12-18 years old at the time of assessment of outcomes”.

The authors replied that: “All studies chosen for inclusion needed to include at least some individuals within the stipulated 12-18 year old bracket, as per our inclusion criteria.”

I understand this reply and I agree it is a reasonable approach. However, several of the included studies explicitly excluded participants below age 18, and included participants age 18+ (as presented by the authors in Table 1). Hence, those studies must be removed from the review as they did not include or overlap with the 12-18 age range.

In their reply, authors argued that studies have used different age ranges to study groups such as ‘young people’. This may be the case. However, if the review stipulates an inclusion criterium of ages 12-18, it seems reasonable to expect that the review adheres to its inclusion criterium.

7. PLOS authors have the option to publish the peer review history of their article (what does this mean?). If published, this will include your full peer review and any attached files.

Reviewer #1: No

Reviewer #2: No

Reviewer #3: No

---

## [Author Response · Author response to Decision Letter 1]

16 Jun 2021

Response to Reviewers

Reviewer #1: I have considered the revisions made by the author and although it is clear that a lot of work went into this review the fundamental design limitations have not been properly addressed.

As I noted in my previous review that heterogeneity of this kind has and can be addressed through meta-analysis, in the very field that this review sits. It is important that we continue to move the field forward, building upon existing evidence. Not doing so, adds further noise to a field that has on more than one occasion suffered from spurious evidence. This is particularly the case when the authors have sought to answer what is essentially a quantitative question (Is suicidal behaviour unique to those who are bereaved by suicide vs other causes of death) with a narrative synthesis. The methods used in this review do not align with this type of research question and I stand by my previous comment that comparison to individuals bereaved by other causes of death do not inform whether the outcomes are unique to those bereaved by suicide. This research question alone has issues since most risk factors associated with suicidal behaviour have low specificity (i.e unique to suicidal behaviour in the first place.

Response: 

We agree that meta-analysis is a very valuable tool where appropriate. In the present manuscript, however, heterogeneous clinical characteristics (varied populations, geography), study designs (which involved different sources of bias and confounding), and exposures and outcomes measured, suggest that summary measures are not necessarily appropriate here. This has been cautioned by several authors (e.g. Borenstein, Hedges, Higgins, & Rothstein, 2009; Greco, Zangrillo, Biondi-Zoccai, & Landoni, 2013; Haidich, 2010) who highlight that the inclusion of low quality studies in a meta-analytic review, of which there are several in our manuscript, can result in biased and misleading summary measures. 

Indeed, other reviews in this area also refrained from conducting pooled analyses (e.g. Andriessen, Draper, Dudley, & Mitchell, 2016; Hua, Bugeja, & Maple, 2019; Kuramoto, Brent, & Wilcox, 2009; Pitman, Osborn, King, & Erlangsen, 2014; Sveen & Walby, 2008) on the basis of variability in outcome measures, study populations, kinship relationships, and study designs, as well as concerns about the potential for confounding and selection biases to impact pooled results.

The publication by Hill et al. (2020) which offers meta-analytic findings focuses on a different study population (i.e. does not compare those exposed to suicide with those exposed to non-suicide death comparison groups), and thus faces different considerations with regards to heterogeneity. Other meta-analyses in the wider field (e.g. Witt et al., 2019) do not use similar exposure/comparison groups, and some only conduct meta-analysis on data from randomised controlled trials (e.g. Robinson et al., 2018). For these reasons, we feel justified in presenting a narrative summary of data from a strictly conducted systematic review which follows our study protocol developed a priori. 

We disagree that comparison with individuals bereaved by other causes does not inform outcomes unique to those bereaved by suicide. A similar approach was taken by Sveen and Walby (2008) and Pitman et al. (2014), who also stressed that comparison with those bereaved by other causes allows for an examination of unique outcomes to suicide bereavement, rather than factors associated with the experience of bereavement in general. 

We agree that bereavement by suicide would represent a risk factor with low specificity if found to be associated with self-harm or suicide-related outcomes. However, we still feel this merits further investigation, and certainly advances research in this field. It is acknowledged that research must look beyond the study of individual risk factors in isolation to predict self-harm or suicide outcomes (e.g. O'Connor & Kirtley, 2018). For this reason, we support the application of theory in research which considers the wider biopsychosocial context of individuals to better understand the complex pathways to suicide. Nevertheless, research on specific factors (such as suicide bereavement) is still beneficial to help capture this broader context of suicide. 

Reviewer #2: Systematic review of adolescent suicide article REVISION

Although in my estimation this article submission has not been fully cleansed of all its minor and major imperfections, it is sufficiently on target to be of value to those attempting to make sense and coherent meaning from the large body of research out there on the effects of suicide bereavement upon adolescents, and will be of value to researchers and clinicians. The authors have worked diligently and effectively to address the criticisms offered by all reviewers and have responded satisfactorily to my own criticisms. I think there is one essential rewording required for the study abstract to make the study’s contents clearer. The first sentence of the methods section of the abstract should also include these words: “including several that extended beyond this age range.” With this amendment, I recommend acceptance of this manuscript for publication.

Response: We appreciate the reviewer’s comments and are pleased that they consider the manuscript worthy of acceptance for publication with their suggested amendment. In light of the changes we have now made to the manuscript following the feedback from Reviewer #3, we hope that these amendments address the reviewer’s concerns regarding the methods section of the abstract. 

Reviewer #3: Dear authors thank you for your replies and for addressing my concerns. While I acknowledge that the manuscript has improved, I notice that one crucial concern has not been addressed. In my review I highlighted that several of the selected studies did not match with the age range criterium that was stipulated in the inclusion criteria. According to the inclusion criteria, selected studies “needed to include adolescent participants, who were defined as being between the ages of 12-18 years old at the time of assessment of outcomes”.

The authors replied that: “All studies chosen for inclusion needed to include at least some individuals within the stipulated 12-18 year old bracket, as per our inclusion criteria.”

I understand this reply and I agree it is a reasonable approach. However, several of the included studies explicitly excluded participants below age 18, and included participants age 18+ (as presented by the authors in Table 1). Hence, those studies must be removed from the review as they did not include or overlap with the 12-18 age range.

In their reply, authors argued that studies have used different age ranges to study groups such as ‘young people’. This may be the case. However, if the review stipulates an inclusion criterium of ages 12-18, it seems reasonable to expect that the review adheres to its inclusion criterium.

Response: We appreciate the reviewer’s comments on this issue. We had initially deemed that by including studies which sampled individuals aged 18 and above, this did in fact meet our inclusion criteria of 12 to 18 year olds (at least for a small number of participants in these studies). However, we have given much thought to the reviewer’s comments, and weighed our concerns regarding the applicability of the findings to adolescent populations, against the risk of loss of data regarding young people if we excluded these studies. 

After much consideration, we feel that although studies which involve 18 year olds and above may represent the experience of some adolescents, we agree that the nine papers in question here which only sampled 18+ year olds focus primarily on the experience of older adults. For this reason, we are in agreement with the reviewer that the manuscript will be strengthened by excluding these studies, and offers more convincing evidence of the adolescent experience of bereavement following suicide or other deaths. 

Following a review of the data after the removal of these nine publications, we can report that our conclusions across all outcome categories do not change materially with the exclusion of these studies. We now present an updated manuscript with amended tables, flow chart, and results sections, as well as comments added to the methods and discussion sections to reflect this change. We thank the reviewer for their feedback and feel this has improved the manuscript considerably. 

 

References

Andriessen, K., Draper, B., Dudley, M., & Mitchell, P. B. (2016). Pre- and postloss features of adolescent suicide bereavement: A systematic review. Death Studies, 40(4), 229-246. doi:10.1080/07481187.2015.1128497

Borenstein, M., Hedges, L. V., Higgins, J. P. T., & Rothstein, H. R. (2009). When Does it Make Sense to Perform a Meta-Analysis? Introduction to Meta-Analysis: John Wiley & Sons, Ltd.

Greco, T., Zangrillo, A., Biondi-Zoccai, G., & Landoni, G. (2013). Meta-analysis: pitfalls and hints. Heart, Lung and Vessels, 5(4), 219-225. 

Haidich, A. B. (2010). Meta-analysis in medical research. Hippokratia, 14(Suppl 1), 29-37. 

Hill, N. T. M., Robinson, J., Pirkis, J., Andriessen, K., Krysinska, K., Payne, A., . . . Lampit, A. (2020). Association of suicidal behavior with exposure to suicide and suicide attempt: A systematic review and multilevel meta-analysis. PLoS Med, 17(3), e1003074. doi:10.1371/journal.pmed.1003074

Hua, P., Bugeja, L., & Maple, M. (2019). A systematic review on the relationship between childhood exposure to external cause parental death, including suicide, on subsequent suicidal behaviour. J Affect Disord, 257, 723-734. doi:10.1016/j.jad.2019.07.082

Kuramoto, S. J., Brent, D. A., & Wilcox, H. C. (2009). The impact of parental suicide on child and adolescent offspring. Suicide and Life-Threatening Behavior, 39(2), 137-151. 

O'Connor, R. C., & Kirtley, O. J. (2018). The integrated motivational-volitional model of suicidal behaviour. Philosophical Transactions of the Royal Society B, 373(1754), 1-10. doi:10.1098/rstb.2017.0268

Pitman, A., Osborn, D., King, M., & Erlangsen, A. (2014). Effects of suicide bereavement on mental health and suicide risk. The Lancet Psychiatry, 1(1), 86-94. doi:10.1016/s2215-0366(14)70224-x

Robinson, J., Bailey, E., Witt, K., Stefanac, N., Milner, A., Currier, D., . . . Hetrick, S. (2018). What Works in Youth Suicide Prevention? A Systematic Review and Meta-Analysis. EClinicalMedicine, 4-5, 52-91. doi:10.1016/j.eclinm.2018.10.004

Sveen, C.-A., & Walby, F. A. (2008). Suicide survivors' mental health and grief reactions: A systematic review of controlled studies. Suicide and Life-Threatening Behavior, 38(1), 13-29. 

Witt, K., Milner, A., Spittal, M. J., Hetrick, S., Robinson, J., Pirkis, J., & Carter, G. (2019). Population attributable risk of factors associated with the repetition of self-harm behaviour in young people presenting to clinical services: a systematic review and meta-analysis. European Child & Adolescent Psychiatry, 28(1), 5-18. doi:10.1007/s00787-018-1111-6

---

## [Editor Report · Decision Letter 2]

23 Jun 2021

A systematic review of controlled studies of suicidal and self-harming behaviours in adolescents following bereavement by suicide

PONE-D-20-33234R2

Dear Dr. Rasmussen,

We’re pleased to inform you that your manuscript has been judged scientifically suitable for publication and will be formally accepted for publication once it meets all outstanding technical requirements.

Kind regards,

Michelle Tye, Ph.D.

Academic Editor

PLOS ONE
---

## [Editor Report · Acceptance letter]

28 Jun 2021

PONE-D-20-33234R2 

A systematic review of controlled studies of suicidal and self-harming behaviours in adolescents following bereavement by suicide 

Dear Dr. Rasmussen:

I'm pleased to inform you that your manuscript has been deemed suitable for publication in PLOS ONE. Congratulations! Your manuscript is now with our production department. 

Kind regards, 

on behalf of

Dr. Michelle Tye 

Academic Editor

PLOS ONE